# Aging is associated with functional and molecular changes in distinct hematopoietic stem cell subsets

Tsu-Yi Su [ORCID] [1,2,7], Julia Hauenstein [ORCID] [3,7], Ece Somuncular [1,2,7], Özge Dumral [ORCID] [1,2], Elory Leonard [1,2], Charlotte Gustafsson [3], Efthymios Tzortzis [ORCID] [1,2], Aurora Forlani [1,2], Anne-Sofie Johansson [ORCID] [1,2], Hong Qian [ORCID] [1,2,4], Robert Månsson [3,5,6] & Sidinh Luc [ORCID] [1,2,4] [✉]

Age is a risk factor for hematologic malignancies. Attributes of the aging hematopoietic system include increased myelopoiesis, impaired adaptive immunity, and a functional decline of the hematopoietic stem cells (HSCs) that maintain hematopoiesis. Changes in the composition of diverse HSC subsets have been suggested to be responsible for age-related alterations, however, the underlying regulatory mechanisms are incompletely understood in the context of HSC heterogeneity. In this study, we investigated how distinct HSC subsets, separated by CD49b, functionally and molecularly change their behavior with age. We demonstrate that the lineage differentiation of both lymphoid-biased and myeloid-biased HSC subsets progressively shifts to a higher myeloid cellular output during aging. In parallel, we show that HSCs selectively undergo age-dependent gene expression and gene regulatory changes in a progressive manner, which is initiated already in the juvenile stage. Overall, our studies suggest that aging intrinsically alters both cellular and molecular properties of HSCs.

Aging of an organism is associated with physiological changes across all organ systems and a progressive functional decline. The age-related functional impairment leads to difficulties in maintaining homeostasis, particularly during stress. Consequently, aging has many health implications and is one of the main risk factors for cancer[1–4]. Physiological aging of the hematopoietic system is associated with decreased competence of the immune system, onset of anemia, myeloid predominance, and increased risks of hematologic disorders[1–3,5]. Throughout life, the entire hematopoietic system is maintained and replenished by hematopoietic stem cells (HSCs)[6]. It has been suggested that aging features of the hematopoietic system are due to functional alterations in the capacity of HSCs to maintain homeostasis. HSCs increase in both frequency and number with age, but the

regenerative capacity of aged HSCs is reduced compared to their young counterparts, indicating that the diminished function is partly counterbalanced by increased HSC numbers to maintain homeostasis[1,2,7,8].

It is well-recognized that the HSC compartment is functionally diverse containing not only lineage-balanced HSCs, but also myeloid-, platelet-, and lymphoid-biased HSC subsets that preferentially generate cells of specific blood lineages[9–13]. Different models have been proposed to underlie the myeloid skewing of the hematopoietic system. In the HSC clonal composition model, where lineage differentiation potential of individual HSCs remains unchanged, the aging-related myeloid predominance is attributed to an increase in platelet- and myeloid-biased HSCs, with a decrease in lymphoid-biased HSCs[7,11,14–17].

[1]Center for Hematology and Regenerative Medicine, Stockholm, Sweden. [2]Department of Medicine Huddinge, Karolinska Institutet, Stockholm, Sweden. [3]Department of Laboratory Medicine, Karolinska Institutet, Stockholm, Sweden. [4]Hematology Center, Karolinska University Hospital, Stockholm, Sweden. [5]Department of Clinical Immunology and Transfusion Medicine, Karolinska University Hospital, Stockholm, Sweden. [6]Science for Life Laboratory, KTH Royal Institute of Technology, Stockholm, Sweden. [7]These authors contributed equally: Tsu-Yi Su, Julia Hauenstein, Ece Somuncular. [✉]e-mail: Sidinh.Luc@ki.se

Conversely, in the cell-intrinsic model, changes in the differentiation properties of HSCs result in diminished ability to generate lymphoid cells, leading to an accumulation of myeloid cells[18,19]. Further studies are needed to elucidate whether distinct HSC subsets undergo age-dependent intrinsic functional changes that underlie the myeloid bias of the aging hematopoietic system[1,2,19].

Aging is accompanied with extensive transcriptional and epigenetic alterations associated with HSC proliferation and differentiation. Comprehensive epigenome studies have shown that gene loci associated with differentiation are hypermethylated, while loci correlated with HSC self-renewal are hypomethylated and display an increase in activating histone marks in aged HSCs[18,20–22]. To what degree these molecular differences reflect the changing composition of functionally different HSC subsets in aging has not been elucidated. Epigenetic characterization of highly enriched lineage-biased HSC subsets has thus far not been widely performed due to limitations in prospectively isolating functionally distinct HSCs. Consequently, epigenetic changes associated with lineage-biased HSCs in aging remain largely unexplored.

We have previously used the integrin CD49b as a prospective marker to distinguish functionally different subsets within the primitive Lineage⁻Sca-1⁺c-Kit⁺ (LSK) CD48⁻CD34⁻CD150ʰⁱ (CD150ʰⁱ) HSC compartment[23]. We demonstrated that the CD49b⁻ subset is highly enriched in myeloid-biased cells, while the CD49b⁺ fraction mainly showed lymphoid-biased features. Furthermore, we showed that CD49b⁻ and CD49b⁺ HSCs were transcriptionally similar but had distinct chromatin accessibility profiles, suggesting that functional differences between lineage-biased HSCs are epigenetically regulated[23].

In the present study, we assessed the functional and molecular changes of distinct HSC subsets phenotypically separated by CD49b in juvenile[24], adult, and old mice. We found that cell proliferation and cell cycle kinetics dynamically change, with increased in vivo myelopoiesis from both CD49b HSC subsets with age. Molecular characterization revealed age-dependent transcriptional and epigenetic changes that preferentially occurred in HSCs. Our studies demonstrate that aging is associated with progressive functional and molecular changes in both CD49b⁻ and CD49b⁺ HSCs, including altered blood lineage output, gene expression, and remodeling of the chromatin landscape.

## Results

### CD49b expression in the HSC compartment is conserved in aging

With age, there is an increased number of total bone marrow (BM) cells, myeloid cells, and phenotypic HSCs[1]. Since CD49b can subfractionate HSCs into lineage-biased subsets[23], we examined whether the frequency of CD49b⁻ and CD49b⁺ cells alters with age. Given that HSCs acquire an adult phenotype at 3–4 weeks after birth[25], we investigated the phenotypic CD150ʰⁱ HSC compartment in juvenile (~1 month), adult (~2–4 months), and old (~1.5–2 years) mice. The CD150ʰⁱ fraction was significantly expanded with age, as previously reported (Fig. 1a and Supplementary Fig. 1a)[14]. Furthermore, the total frequency and number of CD49b⁻ and CD49b⁺ cells increased in old mice, compared to juvenile and adult mice (Fig. 1b, c). However, the distribution pattern of CD49b⁻ and CD49b⁺ subfractions within the CD150ʰⁱ compartment was similar in all ages (Supplementary Fig. 1b, c).

HSCs largely reside in a quiescent state[26]. We previously demonstrated that CD49b⁻ HSCs are more quiescent than CD49b⁺ HSCs in adult mice[23]. To investigate cell cycle changes in the HSC subsets throughout aging, we performed cell cycle analysis using Ki-67 staining (Supplementary Fig. 1d). Cells from both CD49b subsets became progressively more quiescent (G0 phase) with a corresponding decreased G1 fraction with age (Fig. 1d). Furthermore, CD49b⁻ HSCs from adult and old mice were significantly more quiescent than their CD49b⁺ counterparts, unlike in juvenile mice (Supplementary Fig. 1e).

Cell division analysis demonstrated a significantly higher proportion of juvenile CD49b⁻ and CD49b⁺ cells undergoing cell division compared to their adult and old counterparts, compatible with less cells in G0 (Fig. 1e). In vivo labeling with 5-bromo-2'-deoxyuridine (BrdU) revealed dramatically decreased proliferation of both CD49b subsets with age (Fig. 1f and Supplementary Fig. 1f). Furthermore, CD49b⁺ subsets are more proliferative than their corresponding CD49b⁻ cells (Supplementary Fig. 1g). Altogether, our results show that CD49b⁻ and CD49b⁺ cells become more quiescent with age, with the CD49b⁻ HSCs being the more dormant subset.

### The lineage repopulation patterns of multipotent CD49b subsets change with age

The differentiation potential of HSCs has been suggested to alter with age[1,7,18]. We therefore investigated the lymphoid, myeloid, and megakaryocyte potential of CD49b⁻ and CD49b⁺ subsets clonally in vitro. In all age groups, both CD49b subsets generated B lymphocytes and myeloid cells (Fig. 2a). Consistent with previous results, the HSC subsets primarily generated myeloid cells, and less B cells in OP9 co-cultures[23]. Furthermore, CD49b⁻ and CD49b⁺ cells from all ages efficiently generated megakaryocytes with no significant differences (Fig. 2b). Thus, the lineage differentiation ability in vitro is preserved with age.

To evaluate age-associated differences in the differentiation ability of CD49b subsets in vivo, we performed competitive transplantation experiments using the *Gata-1* eGFP mouse strain to permit the detection of platelets and erythrocytes, in addition to leukocytes[27]. A limiting dose of five cells from juvenile or adult CD49b⁻ and CD49b⁺ subsets were transplanted into each recipient, while one hundred cells from old mice were transplanted to account for the reduced regenerative capacity of old HSCs[1,2,7,8]. As expected, the total leukocyte contribution of old HSC subsets was comparable to their juvenile and adult counterparts (Fig. 2c). Although both CD49b subsets across all ages demonstrated long-term (LT) multilineage repopulating ability, CD49b⁺ cells preferentially repopulated lymphoid cells in all age groups (Fig. 2d and Supplementary Figs. 2 and 3a–c). To determine the lineage distribution, we analyzed the relative contribution of lymphoid and myeloid cells within the donor leukocyte compartment of individually transplanted mice (Fig. 2e). Using peripheral blood (PB) profiles from adult unmanipulated mice as reference (Supplementary Fig. 3d), the lineage distribution of transplanted mice was categorized based on the lymphoid to myeloid blood cell (L/M) ratio (Supplementary Fig. 3e)[23,28]. Consistent with previous data[23], the most frequent categorization in juvenile and adult CD49b⁻ cells was the myeloid-biased (M-bi) pattern, whereas lymphoid-biased (L-bi) was most common in CD49b⁺ cells (Fig. 2f). In contrast to the juvenile and adult age groups, both CD49b subsets in old mice were highly M-bi (Fig. 2f). Given the overall higher myeloid contribution in old mice, we used the L/M ratio from old unmanipulated mice as a reference to assess whether old CD49b⁻ and CD49b⁺ transplanted mice enriched for different lineage distribution patterns (Supplementary Fig. 3e). Indeed, old CD49b⁺ HSCs greatly enriched for cells with a balanced (Bal) cellular output, while M-bi was still the predominant classification in old CD49b⁻ HSCs (Fig. 2g and Supplementary Fig. 3f). Consistent with blood repopulation patterns but unlike their juvenile counterparts, both old CD49b subsets had higher relative myeloid repopulation in the BM (Fig. 2h). Furthermore, nearly all mice transplanted with old CD49b⁻ and CD49b⁺ HSCs repopulated the granulocyte-monocyte progenitors (GMPs) and megakaryocyte progenitors (MkPs), whereas only a few mice reconstituted the common lymphoid progenitors (CLPs) and lymphoid-primed multipotent progenitors (LMPPs) (Supplementary Fig. 4a–d). We also observed that CD41 expression increased in both CD49b⁻ and CD49b⁺ HSCs with age, consistent with CD41 marking myeloid-biased cells (Supplementary Fig. 4e)[29].

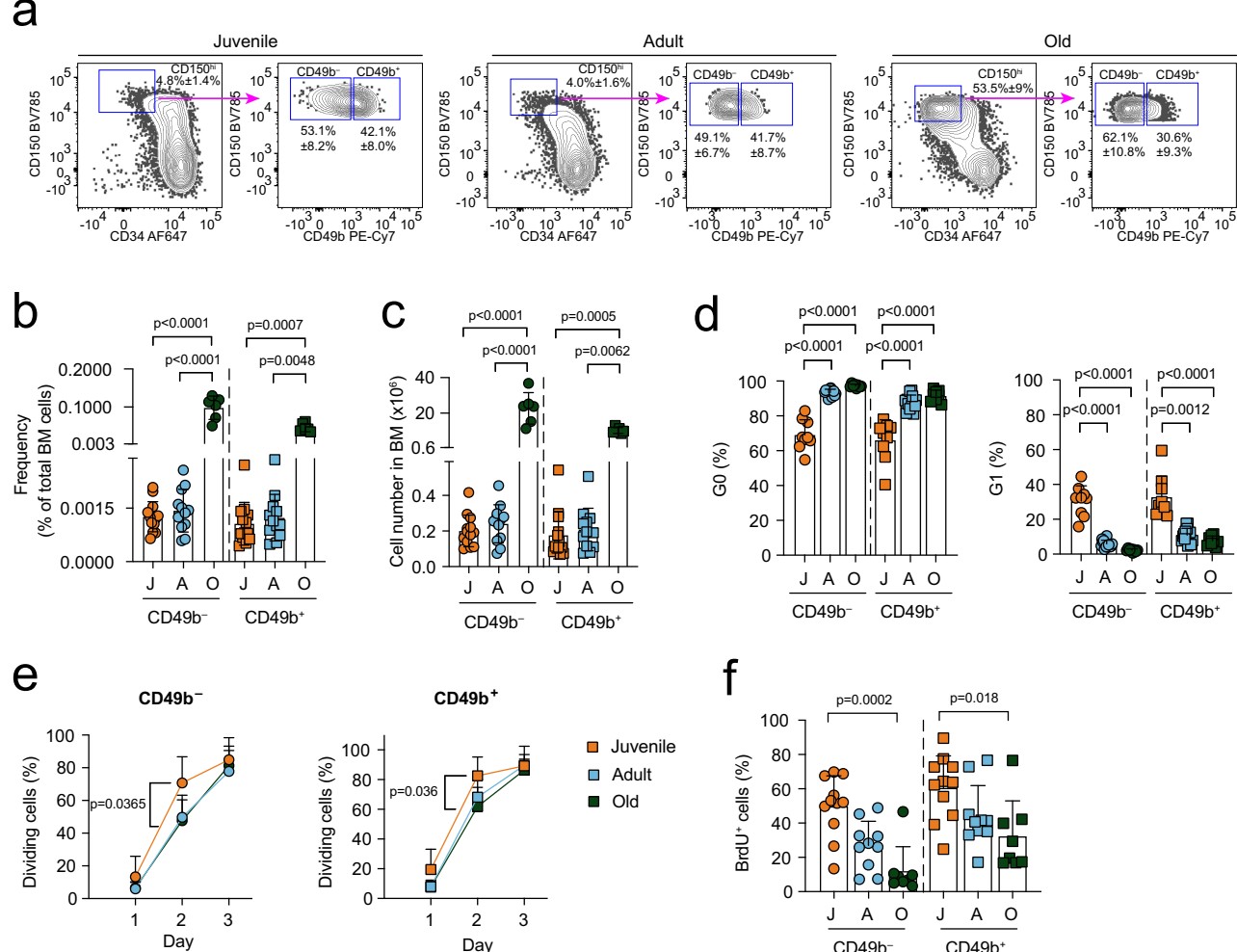

**Fig. 1 | CD49b expression in the HSC compartment is conserved in aging. a** FACS profiles and gating strategy of the phenotypic HSC compartment (Lineage⁻Sca-1⁺c-Kit⁺ (LSK) CD48⁻CD34⁻CD150ʰⁱ), with further separation using CD49b, in juvenile, adult, and old mice. Frequency of parent gates are shown. See Supplementary Fig. 1a for the full gating strategy. **b** Total frequency of CD49b⁻ and CD49b⁺ HSC subsets in juvenile ($n = 13$ mice, 3 experiments), adult ($n = 12$ mice, 6 experiments), and old ($n = 6$ mice, 6 experiments) mice. **c** Total numbers of CD49b⁻ and CD49b⁺ HSC subsets in juvenile ($n = 13$ mice, 3 experiments), adult ($n = 12$ mice, 6 experiments), and old ($n = 6$ mice, 6 experiments) mice. **d** Frequency of CD49b⁻ and CD49b⁺ HSCs in G0 (left) and G1 (right) of juvenile ($n = 9$ mice, 3 experiments), adult ($n = 15$ mice, 6 experiments), and old ($n = 8$ mice, 5 experiments) mice. **e** Frequency of cell divisions from cultured single cells of CD49b⁻ (left) and CD49b⁺ (right) HSCs

at days 1–3 from juvenile ($n = 7$ mice, 3 experiments, $n_{CD49b^-} = 351$ cells, $n_{CD49b^+} = 258$ cells), adult ($n = 4$ mice, 3 experiments, $n_{CD49b^-} = 149$ cells, $n_{CD49b^+} = 146$ cells), and old ($n = 6$ mice, 4 experiments, $n_{CD49b^-} = 599$ cells, $n_{CD49b^+} = 539$ cells) mice. **f** Frequency of BrdU⁺ CD49b⁻ and CD49b⁺ HSCs from juvenile ($n = 11$ mice, 3 experiments), adult ($n = 10$ mice, 3 experiments), and old ($n = 8$ mice, 3 experiments) mice. Mean ± s.d. is shown. The statistical analyses were performed two-sided with one-way ANOVA with Tukey's multiple comparison test in the CD49b⁻ subset in **b**–**d** and CD49b⁺ G0 in **d**, two-way repeated measures ANOVA with Tukey's multiple comparison test in **e**, Kruskal–Wallis with Dunn's multiple comparison test in the CD49b⁺ subset in **b** and **c**, CD49b⁺ G1 in **d**, and in **f**. J juvenile, A adult, O old. See also Supplementary Fig. 1. Source data are provided as a Source Data file.

Our findings demonstrate that CD49b distinguishes functionally different HSC subsets, and that both CD49b⁻ and CD49b⁺ cells increase their myeloid contribution with age.

## The CD49b⁻ HSC subset is the most durable subset regardless of age

We have previously shown that adult CD49b subsets differ in their durable self-renewal potential[23]. To investigate the changes in extensive self-renewal abilities of CD49b⁻ and CD49b⁺ subsets during aging, we assessed the number of mice exhibiting LT myeloid repopulation in the blood, as a measure of ongoing HSC activity. LT myeloid repopulation was more frequently found in mice transplanted with juvenile CD49b⁻ compared to CD49b⁺ cells, which was substantiated by the significantly higher number of mice reconstituting phenotypic HSCs in primary transplantation (Fig. 3a). However, they showed similar overall

repopulation level and regenerated both phenotypic CD49b HSC subsets (Fig. 3b and Supplementary Fig. 4f). In contrast, all reconstituted mice from the old age group had LT myeloid repopulation in primary transplantation, compatible with the high frequency of mice that also repopulated phenotypic HSCs, likely reflecting the larger number of transplanted cells (Fig. 3c). CD49b populations from old mice also regenerated both CD49b HSCs subtypes (Fig. 3b and Supplementary Fig. 4g). To examine the self-renewal potential of juvenile and old CD49b subsets more conclusively, transplanted mice exhibiting phenotypic HSC repopulation were secondary transplanted. The repopulation patterns were generally preserved through serial transplantation (Supplementary Fig. 4h,i), as expected[23,28]. Although the total leukocyte contribution was comparable for juvenile and old HSC subsets, the CD49b⁺ subsets exhibited significantly less repopulation (Fig. 3d). Most mice from juvenile CD49b populations exhibited LT

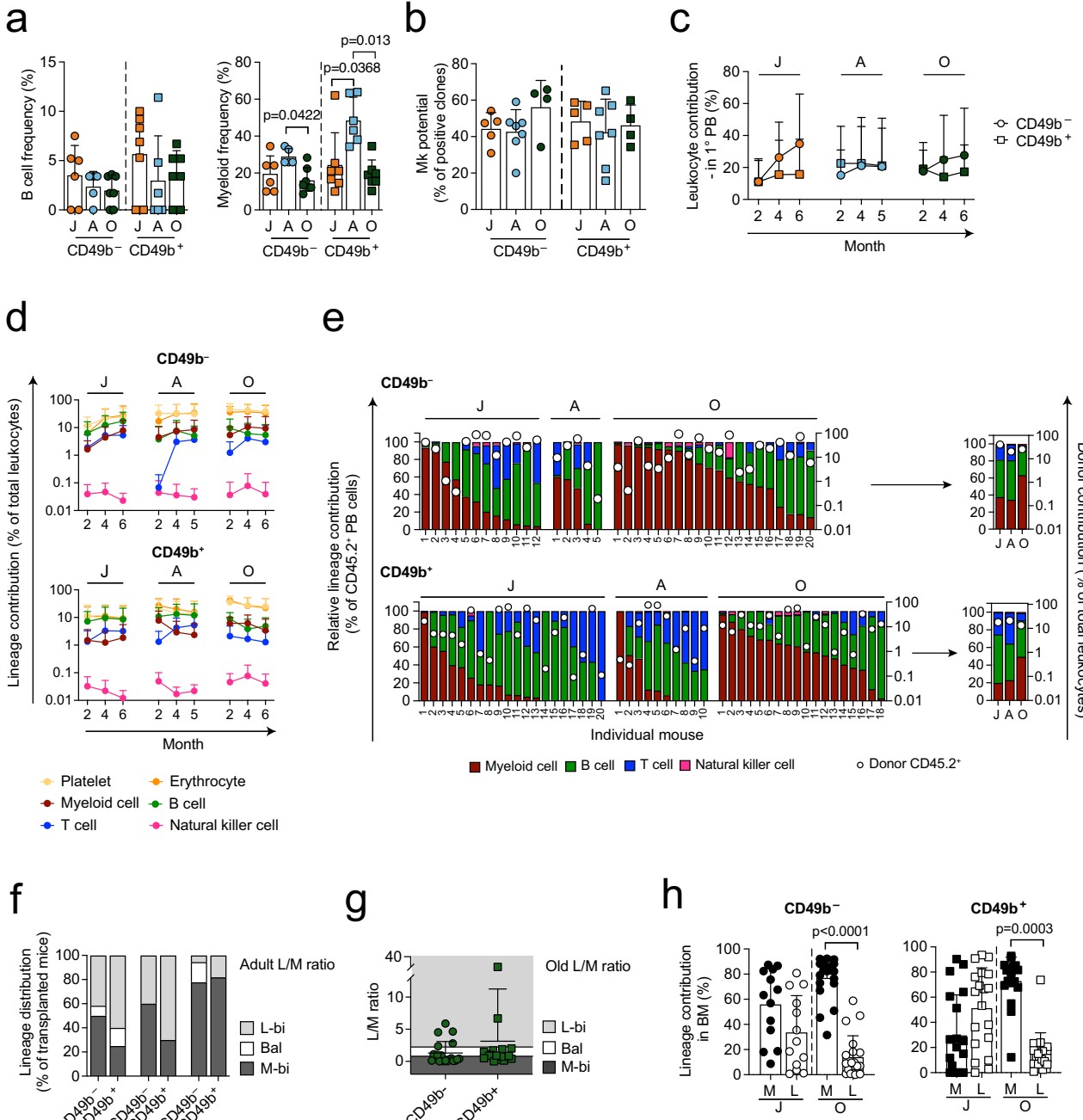

**Fig. 2 | The lineage repopulation patterns of multipotent CD49b subsets change with age. a** Total frequency of clones containing B cells (left), or only myeloid cells (right) from CD49b subsets in juvenile ($n_{CD49b^-}$ = 6 mice, $n_{CD49b^+}$ = 7 mice, 3 experiments), adult ($n_{CD49b^-}$ = 5 mice, $n_{CD49b^+}$ = 6 mice, 3 experiments), and old ($n_{CD49b^-}$ = 7 mice, $n_{CD49b^+}$ = 7 mice, 4 experiments) mice. **b** Megakaryocyte potential of CD49b subsets from juvenile ($n_{CD49b^-}$ = 5 mice, $n_{CD49b^+}$ = 5 mice, 3 experiments), adult ($n_{CD49b^-}$ = 7 mice, $n_{CD49b^+}$ = 7 mice, 6 experiments), and old ($n_{CD49b^-}$ = 4 mice, $n_{CD49b^+}$ = 4 mice, 4 experiments) mice. **c** Donor leukocyte contribution in the PB of transplanted mice ($n^I_{CD49b^-}$ = 14 mice, $n^I_{CD49b^+}$ = 24 mice, $n^A_{CD49b^-}$ = 5 mice, $n^A_{CD49b^+}$ = 10 mice, $n^O_{CD49b^-}$ = 20 mice, $n^O_{CD49b^+}$ = 18 mice). **d** Donor-derived lineage contribution in the PB of transplanted mice ($n^I_{CD49b^-}$ = 14 mice, $n^I_{CD49b^+}$ = 24 mice, $n^A_{CD49b^-}$ = 5 mice, $n^A_{CD49b^+}$ = 10 mice, $n^O_{CD49b^-}$ = 20 mice (11 for P-E), and $n^O_{CD49b^+}$ = 18 mice (15 for P-E)). **e** Relative lineage contribution within donor leukocytes and donor chimerism (CD45.2) in the PB 5–6 months post-transplantation from **d**. **f** Proportion of lineage distribution patterns from **e**, using

adult L/M ratio ($n^I_{CD49b^-}$ = 12 mice, $n^I_{CD49b^+}$ = 20 mice, $n^A_{CD49b^-}$ = 5 mice, $n^A_{CD49b^+}$ = 10 mice, $n^O_{CD49b^-}$ = 20 mice, $n^O_{CD49b^+}$ = 18 mice). **g** L/M ratio in PB of mice transplanted with old CD49b subsets ($n_{CD49b^-}$ = 20 mice, $n_{CD49b^+}$ = 18 mice) 5–6 months post-transplantation. The ranges for L-bi, Bal, and M-bi based on old L/M ratio are indicated. **h** Relative lineage contribution to myeloid (M) and lymphoid cells (L: B, T, and NK cells) in the BM of transplanted mice ($n^I_{CD49b^-}$ = 13 mice, $n^I_{CD49b^+}$ = 19 mice, $n^O_{CD49b^-}$ = 20 mice, $n^O_{CD49b^+}$ = 18 mice). Mean ± s.d. is shown. Statistical analyses were performed two-sided, with Kruskal–Wallis with Dunn's multiple comparison test in **a** and CD49b⁻ in **b**, one-way ANOVA with Tukey's multiple comparison test in CD49b⁺ in **b**, Mann–Whitney test in **c**, and Wilcoxon signed-rank test in **h**, except in J$_{CD49b^-}$, where paired t-test was done. J juvenile, A adult, O old, PB peripheral blood, BM bone marrow, L-bi lymphoid-biased, Bal balanced, M-bi myeloid-biased, L/M lymphoid to myeloid. See also Supplementary Figs. 2–4. Source data are provided as a Source Data file.

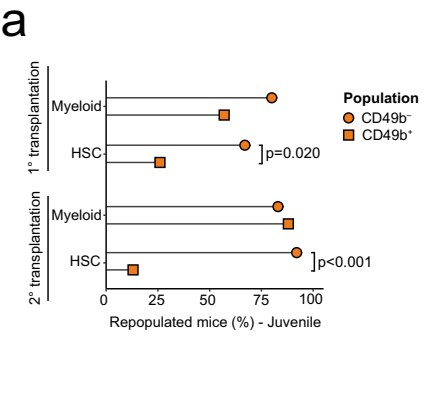

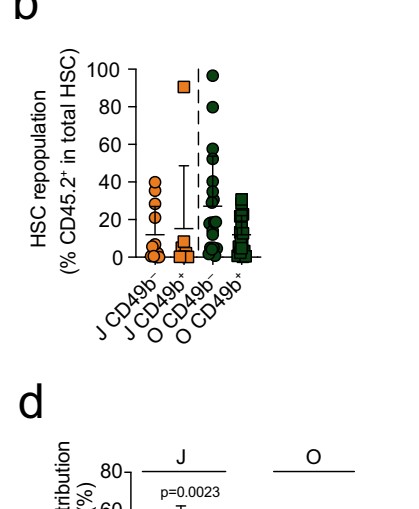

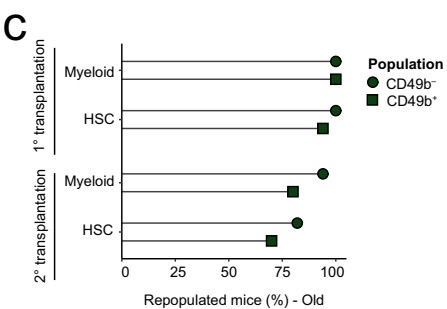

**Fig. 3 | The CD49b⁻ HSC subset is the most durable subset regardless of age.**
**a** Proportion of mice exhibiting myeloid repopulation in the peripheral blood and HSC (LSK CD48⁻Flt-3⁻CD150⁺) repopulation in the bone marrow (BM), 5–6 months after primary or secondary transplantation of juvenile CD49b⁻ and CD49b⁺ HSCs ($n^{CD49b⁻}_{Primary}$ = 15 mice, $n^{CD49b⁺}_{Primary}$ = 23 mice, $n^{CD49b⁻}_{Secondary}$ = 12 mice, $n^{CD49b⁺}_{Secondary}$ = 8 mice). **b** Frequency of HSC repopulation in reconstituted mice after primary transplantation ($n^{J}_{CD49b⁻}$ = 12 mice, $n^{J}_{CD49b⁺}$ = 7 mice, $n^{O}_{CD49b⁻}$ = 20 mice, $n^{O}_{CD49b⁺}$ = 17 mice). **c** Proportion of mice exhibiting myeloid repopulation in the peripheral blood and HSC (LSK CD48⁻Flt-3⁻CD150⁺) repopulation in the BM, 5–6 months after primary or secondary transplantation of old CD49b⁻ and CD49b⁺

HSCs ($n^{CD49b⁻}_{Primary}$ = 20 mice, $n^{CD49b⁺}_{Primary}$ = 18 mice, $n^{CD49b⁻}_{Secondary}$ = 17 mice, $n^{CD49b⁺}_{Secondary}$ = 10 mice). **d** Total donor leukocyte contribution in the PB of mice secondary transplanted with CD49b⁻ and CD49b⁺ HSC subsets from juvenile, adult, and old mice ($n^{J}_{CD49b⁻}$ = 12 mice, $n^{J}_{CD49b⁺}$ = 8 mice, $n^{O}_{CD49b⁻}$ = 17 mice, $n^{O}_{CD49b⁺}$ = 10 mice). Mean ± s.d. is shown. The statistical analyses were performed two-sided, with Fisher´s exact test in **a** and **c**, Mann–Whitney test in **b**, and Šídák's multiple comparisons test after adjusting for repeated measures in **d**. J juvenile, O old, PB peripheral blood. See also Supplementary Fig. 4. Source data are provided as a Source Data file.

myeloid repopulation, but a significantly higher number from the CD49b⁻ group reconstituted phenotypic HSCs than the CD49b⁺ subset (Fig. 3a). In contrast, we found higher frequencies of mice reconstituting both myeloid cells and HSCs from old CD49b⁻ cells compared to old CD49b⁺, although the differences were not statistically significant (Fig. 3c). Our findings are compatible with CD49b⁻ HSCs harboring the highest self-renewal potential[23].

### The CD49b⁻ and CD49b⁺ subsets show similar distribution in the BM but distinct migration properties

HSCs reside in specialized anatomical BM microenvironments[30]. To assess whether functional differences of CD49b⁻ and CD49b⁺ subsets were linked to differential anatomical localization, we analyzed the distribution of these subsets in the BM of steady-state mice. We found similar distribution of both CD49b subsets in the central marrow, endosteal, and trabecular bones in all ages (Supplementary Fig. 5a,b). Nevertheless, both subsets increased in the central marrow with age (Supplementary Fig. 5c). These results indicate that differential HSC function is likely not dependent on HSC distribution in the BM.

In a co-culture assay using mouse BM mesenchymal stem cells (MSCs)[31], we observed more transmigrated CD49b⁺ than CD49b⁻ HSCs, 24–48 h after plating (Supplementary Fig. 5d). This was functionally validated by more Cobblestone-Area-Forming Cells (CAFCs) formed by the CD49b⁺ HSCs (Supplementary Fig. 5e). Furthermore, in old mice, the CAFCs from CD49b⁻ HSCs were reduced, suggesting compromised HSC migration potential with age (Supplementary Fig. 5f)[7].

Altogether, although CD49b⁻ and CD49b⁺ HSCs have distinct migratory capacity, they appear to locate in similar BM regions.

### HSCs undergo considerable gene expression changes during aging

To investigate the molecular mechanisms underlying functional differences of HSC subsets in aging, we performed single cell RNA-sequencing (scRNA-seq) on stem- and progenitor cells from juvenile and old mice (CD49b⁻, CD49b⁺, LMPP, and GMP; Supplementary Data 1), and combined this with our published data from adult mice[23].

Using dimensionality reduction, we observed that HSCs, LMPPs, and GMPs formed distinct clusters (Fig. 4a), with expected expression of genes used for phenotypic definition of the populations, including *Slamf1*, *Flt3*, *Cd48*, *Fcgr2b*, and *Fcgr3* (Supplementary Fig. 6a). While the CD49b subsets were indistinguishable, the HSCs segregated into age-specific clusters, in contrast to the progenitors (Fig. 4a). Comparison to previously published scRNA-seq data showed that our HSCs had a clear LT-HSC identity[32], lacked lineage priming[33], and did not exhibit a fetal HSC signature[34] (Supplementary Fig. 6b–d). Furthermore, the age-specific segregation of HSCs was confirmed using published scRNA-seq data sets (Supplementary Fig. 6e)[16,21,35,36].

We subsequently investigated differentially expressed genes between juvenile and old HSCs (CD49b⁻ and CD49b⁺ combined; $p_{adj}$ < 0.01, $log_2FC$ > 1) and found 173 genes upregulated and 455 genes downregulated with age (Fig. 4b and Supplementary Data 1). Several aging-related genes were observed, including *Selp* (P-selectin) and *Aldh1a1* (aldehyde dehydrogenase 1a1; Supplementary Fig. 6a)[20,37,38]. Generally, these age-associated differences were recapitulated in published studies comparing adult and old HSCs (Supplementary Fig. 6f)[16,21,35,36]. Furthermore, gene set enrichment analysis (GSEA)

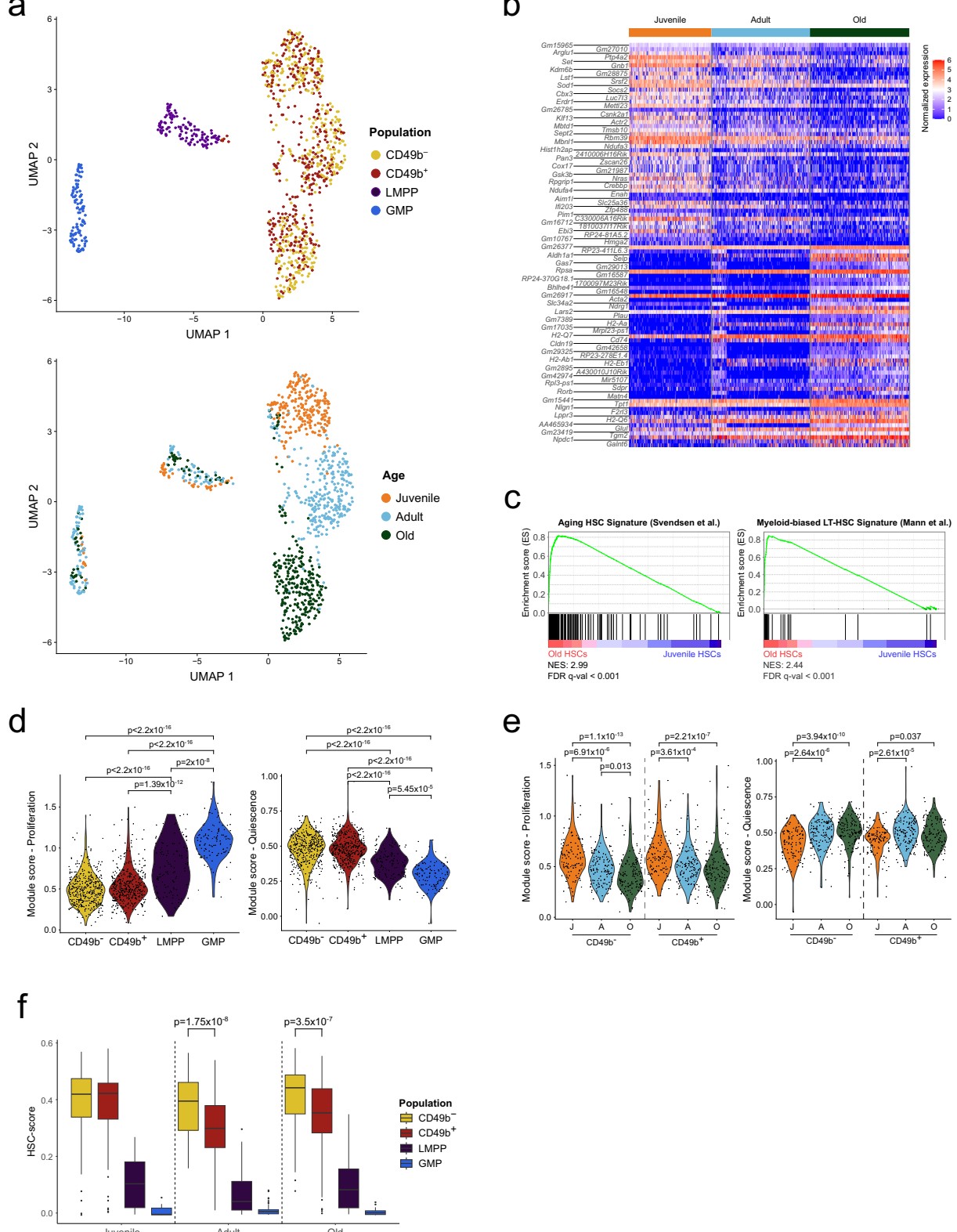

demonstrated that genes associated with HSC aging and myeloid lineage bias had increased expression in old HSCs (Fig. 4c)[35,37].

We demonstrated age-dependent variations in the cell cycle and proliferation analysis of HSCs (Fig. 1d–f). Utilizing published gene sets[39], we found that both CD49b subsets displayed a significantly lower proliferation score and higher quiescence score compared to LMPPs and GMPs (Fig. 4d), as expected. We also observed a

significantly higher proliferation score and lower quiescence score in juvenile HSCs, consistent with the age-related quiescence in old HSCs (Figs. 1d–f and 4e).

We further investigated the molecular mechanism underlying the enhanced migration potential of CD49b+ HSCs (Supplementary Fig. 5d, e). Expression analysis using a gene set of adhesion molecules[40] revealed significant age-associated differences but no subset-related

**Fig. 4 | HSCs undergo considerable gene expression changes during aging.**
**a** UMAP visualization of single-cell RNA-seq data from stem- and progenitor cells, from juvenile, adult, and old mice ($n^J_{CD49b^-}$ = 115 cells, $n^J_{CD49b^+}$ = 118 cells, $n^J_{LMPP}$ = 34 cells, $n^J_{GMP}$ = 11 cells, $n^A_{CD49b^-}$ = 133 cells, $n^A_{CD49b^+}$ = 144 cells, $n^A_{LMPP}$ = 55 cells, $n^A_{GMP}$ = 73 cells, $n^O_{CD49b^-}$ = 145 cells, $n^O_{CD49b^+}$ = 135 cells, $n^O_{LMPP}$ = 32 cells, $n^O_{GMP}$ = 26 cells). Cells are colored by population (top) or age (bottom). **b** Heatmap of normalized expression for differentially expressed genes between juvenile and old HSCs. The top 50 differentially expressed genes up in juvenile and old, with $p_{adj} < 0.01$ and $log_2FC > 1$ (LR test), are shown. **c** Gene set enrichment analysis for old compared to juvenile HSCs and the indicated custom gene sets from Svendsen et al.[37] and Mann et al.[35]. **d** Proliferation and quiescence scores for all analyzed stem- and progenitor cells, regardless of age. **e** Proliferation and quiescence scores for juvenile, adult, and old HSC subsets. **f** Calculated HSC-score for stem- and progenitor cells from juvenile, adult, and old mice. All statistical analyses were performed two-sided, with Kruskal−Wallis with Dunn's multiple comparison test in **d** and **e** and Mann−Whitney test in **f**. Only the HSC subsets were included in the statistical analysis in **f**. NES normalized enrichment score, J juvenile, A adult, O old. See also Supplementary Fig. 6. Source data are provided as a Source Data file.

changes (Supplementary Fig. 6g). Furthermore, we found no changes in the expression of *Itgb1*, the integrin beta1 subunit paired with CD49b, as determined by scRNA-seq and FACS (Supplementary Fig. 6h). This indicates that CD49b expression, rather than *Itgb1*, might contribute to the observed age-related changes in HSC migration.

Congruent with the transcriptional similarity between CD49b⁻ and CD49b⁺ cells (Fig. 4a), no genes were significantly different in juvenile and old age groups ($p_{adj}$ <0.05). However, given the distinct functional differences between CD49b subsets, we applied the hscScore method, which utilizes validated HSC gene expression data sets[41] to assess molecular changes through leveraging small but concordant differences in gene expression. Notably, we observed a significantly lower HSC-score in adult and old CD49b⁺ compared to their corresponding CD49b⁻ cells (Fig. 4f), consistent with the reduced self-renewal potential of CD49b⁺ subsets[23]. However, CD49b⁻ and CD49b⁺ cells from juvenile mice exhibited a similar score despite functional differences. Moreover, we did not observe any consistent age-related changes, altogether suggesting that HSC-score may not predict all self-renewal differences (Supplementary Fig. 6i).

Collectively, these results demonstrate that aging is associated with distinct gene expression changes selectively in HSCs. However, global transcriptome analysis cannot resolve functionally distinct HSC subsets, even at the single cell level[23].

## Aging is associated with a progressive increase of chromatin accessibility in HSCs

Given the high transcriptional similarity between CD49b⁻ and CD49b⁺ cells, we used Assay for Transposase-Accessible Chromatin sequencing (ATAC-seq)[23] to examine age-related epigenetic differences in stem- and progenitor cells from juvenile and old mice (Supplementary Data 2 and Supplementary Fig. 7a, b). Our previous data from adult mice were included for comparison[23]. Genes such as *Slamf1*, *Flt3*, and *Fcgr2b* showed expected population-specific chromatin accessibility profiles (Supplementary Fig. 7c). Principal component analysis (PCA) of the ATAC-seq data revealed distinct clusters of LMPPs, GMPs, and HSCs (CD49b⁻ and CD49b⁺; Fig. 5a), concordant with scRNA-seq data. Although HSC subsets overall created one cluster (Fig. 5a), discrete subclusters based on age and CD49b surface expression were discernable in principal components (PC) 1–3 (Fig. 5b, left and Supplementary Fig. 8a) and PC4 (Fig. 5b, right), respectively. Integration of published ATAC-seq data of corresponding populations further supported this clustering (Supplementary Fig. 8b, c)[22].

As the PCA indicated substantial age-related differences in HSCs, we interrogated chromatin accessibility changes between juvenile and old CD49b subsets. We identified 5501 and 3849 significantly differentially accessible regions (DARs; $p_{adj} < 0.0001$) between juvenile and old subsets (CD49b⁻_Juvenile vs CD49b⁻_Old and CD49b⁺_Juvenile vs CD49b⁺_Old) respectively (Fig. 5c and Supplementary Data 2). The majority of DARs constituted age-associated gain in accessibility, including regions of common age-related genes including *Clu* (Clusterin), *Aldh1a1*, and *Cdc42* (Cell division cycle 42; Supplementary Fig. 8d)[37,42]. The accessibility pattern of these DARs showed a similar trend as published data comparing adult and old HSCs (Supplementary Fig. 8e)[22]. Overall, the pattern of age-dependent chromatin

accessibility changes was highly similar in both CD49b subsets and generally initiated already in adult HSCs (Fig. 5c). These findings suggest common aging mechanisms in both subsets. Strikingly, in old HSCs, regions with gained chromatin accessibility were largely acquired de novo, whereas regions with decreased accessibility frequently lost accessibility completely (Fig. 5d). Interestingly, the chromatin accessibility changes in HSCs are reversible, as most age-associated changes were not propagated to downstream progenitors (Supplementary Fig. 8f).

Gene ontology (GO) analysis of DARs with high accessibility in juvenile HSCs (Fig. 5c, cluster 1) was associated with T cell activation and cell signaling (Fig. 5e). Conversely, DARs with high accessibility in old HSCs (Fig. 5c, clusters 2–3) enriched for processes connected to reactive oxygen species (ROS) and NF-kB signaling (Fig. 5e, cluster 2), myeloid cell differentiation, hematopoiesis, and cell number regulation (Fig. 5e, cluster 3). These results align with the age-related functional decline in lymphopoiesis and increase in myelopoiesis[1], which is supported by the reduced L/M blood cell ratio in mice transplanted with old HSCs (Supplementary Fig. 8g). Additionally, the increased NF-kB signaling and ROS accumulation are recognized aging-associated effects[43–46].

We next performed motif enrichment analysis to assess putative transcription factor (TF) binding in aging HSCs. Notably, DARs with increased accessibility in juvenile HSCs (Fig. 5c, cluster 1) predominantly enriched for ETS family transcription factor binding sites (TFBS), of which SPI1 (PU.1) and SPIB were the most significantly enriched (Fig. 5f). In contrast, DARs with increased accessibility in adult and old HSCs (Fig. 5c, clusters 2–3) were enriched for ETS-, bZIP-, IRF-, and RUNT-family TFBS (Fig. 5f). Congruent with the motif enrichment analysis, *Spi1* and *Spib* gene expression was reduced in aging, whereas expression of *Junb*, *Irf1*, and *Runx1* increased (Supplementary Fig. 9a).

We observed a gradual increase in open chromatin with age, with adult HSC subsets exhibiting intermediate chromatin accessibility (Fig. 5c) that was already initiated in the transition from juvenile to adult HSCs. To further investigate the early aging-associated chromatin regions, we assessed the overlap of DARs between juvenile and adult HSCs, and between juvenile and old HSCs (Fig. 5g). We identified 182 age-related DARs that were already significantly changed from juvenile to adult stages ($p_{adj} < 0.0001$). Motif enrichment analysis of these regions showed enrichment of bZIP-, CTF-, and IRF-family TFBS in DARs with gained accessibility in aging. Conversely, SPIB TFBS were enriched in DARs with reduced accessibility with age (Fig. 5h).

Collectively, our findings demonstrate that aging is primarily associated with an HSC-specific and progressive gain of chromatin accessibility in both CD49b subsets that is already initiated in the juvenile-to-adult transition. Furthermore, our data suggest that aging-related chromatin remodeling is linked to age-dependent changes in TF binding.

## Aging and lineage bias are regulated by the same transcription factor families

Although age-related chromatin changes in CD49b⁻ and CD49b⁺ subsets were highly similar (Fig. 5c), the subsets clustered apart within all age groups in the PCA (Fig. 5b). By investigating the chromatin

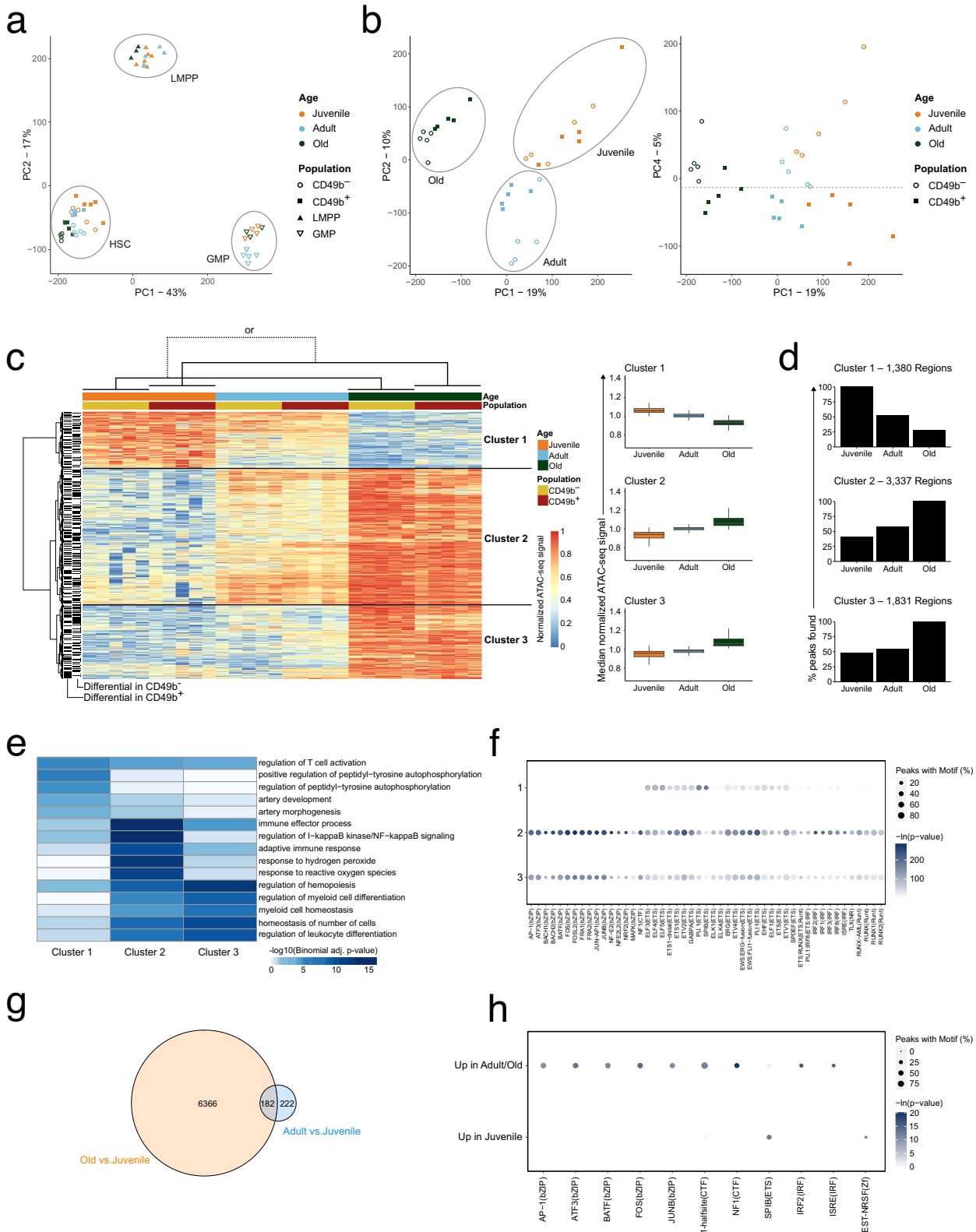

accessibility differences between CD49b⁻ and CD49b⁺ subsets in juvenile and old mice, we identified 161 and 659 DARs ($p_{adj} < 0.05$), respectively (Fig. 6a and Supplementary Data 2). In previously published ATAC-seq data from HSCs without CD49b subfractionation, the identified DARs from old mice displayed intermediate accessibility as anticipated (Supplementary Fig. 9b)[22].

Consistent with the high similarity in chromatin accessibility between CD49b⁻ and CD49b⁺ HSCs in juvenile mice, GO analysis did not yield any significantly enriched terms (Fig. 6b). In contrast, DARs associated with old CD49b⁻ cells enriched for processes including cell cycle and leukocyte regulation, as well as GM-CSF and WNT pathways (Fig. 6b). These findings align with old CD49b⁻ cells being more

**Fig. 5 | Aging is associated with a progressive increase of chromatin accessibility in HSCs. a** Principal component analysis of ATAC-seq data from stem- and progenitor cells, from juvenile, adult, and old mice ($n^J_{CD49b^-}$ = 5 samples, $n^J_{CD49b^+}$ = 5 samples, $n^J_{LMPP}$ = 6 samples, $n^J_{GMP}$ = 4 samples, $n^A_{CD49b^-}$ = 5 samples, $n^A_{CD49b^+}$ = 5 samples, $n^A_{LMPP}$ = 5 samples, $n^A_{GMP}$ = 5 samples, $n^O_{CD49b^-}$ = 5 samples, $n^O_{CD49b^+}$ = 5 samples, $n^O_{LMPP}$ = 3 samples, $n^O_{GMP}$ = 4 samples). **b** Principal component analysis of ATAC-seq data from CD49b$^-$ and CD49b$^+$ HSC subsets, from juvenile, adult, and old mice. ($n^J_{CD49b^-}$ = 5 samples, $n^J_{CD49b^+}$ = 5 samples, $n^A_{CD49b^-}$ = 5 samples, $n^A_{CD49b^+}$ = 5 samples, $n^O_{CD49b^-}$ = 5 samples, $n^O_{CD49b^+}$ = 5 samples). **c** Heatmap (left) of row normalized chromatin accessibility for regions with differential accessibility ($p_{adj}$ <0.0001, Wald test) between juvenile and old CD49b$^-$ and/or between juvenile and old CD49b$^+$ cells. Regions are divided into three clusters based on hierarchical clustering. Median normalized chromatin accessibility of clusters 1–3 is shown (right). **d** Percentage of regions constituting open chromatin in clusters 1–3. **e** Top 5 GO biological processes significantly enriched in clusters 1–3. **f** Transcription factors with enriched binding motifs (-ln(p-value)>50) in clusters 1–3. **g** Venn diagram of regions with differential accessibility ($p_{adj}$ <0.0001, Wald test) in old compared to juvenile HSCs (Old vs. Juvenile) or in adult compared to juvenile HSCs (Adult vs. Juvenile). **h** Transcription factors with enriched binding motifs (-ln(p-value)>10) in regions with increased or decreased accessibility in both adult and old compared to juvenile HSCs. A one-sided binomial test was used to determine significance in **e**, **f**, and **h**. p-values in **c** and **e** were adjusted using the Benjamini–Hochberg method. Boxplots show the distribution in each population (center line, median; box limits, interquartile range; whiskers, furthest data point within 1.5x of the interquartile range). J juvenile, A adult, O old. See also Supplementary Figs. 7–9. Source data are provided as a Source Data file.

quiescent and myeloid-biased compared to CD49b$^+$ cells (Fig. 2g and Supplementary Fig. 1e, g).

In transplantation experiments, we observed considerable differences in blood lineage contribution, with an overall age-related increase in myeloid differentiation in both CD49b subsets. To identify epigenetic changes associated with lineage bias differences, we categorized HSC subsets based on their differentiation characteristics (Fig. 6c). Differential analysis was subsequently performed between the most lymphoid-biased (CD49b$^+_{Juvenile}$) and most myeloid-biased (CD49b$^-_{Old}$) population to identify DARs associated with lineage bias. However, given the substantial chromatin changes in aging HSCs (Fig. 5c, d), we filtered out aging-related DARs (Fig. 5c and Supplementary Data 2) to identify DARs unique to lineage bias (Lin DARs, Fig. 6c, d and Supplementary Data 2). GO analysis revealed enrichment of several blood cell associated processes, including regulation and differentiation of leukocytes, myeloid cells, and neutrophils, confirming that the identified chromatin regions are important for lineage differentiation (Fig. 6e). In further agreement, the Lin DARs included regions associated with the myeloid growth factor receptor, *Csf2ra* (colony stimulating factor 2 receptor, alpha) and hematopoietic regulator, *Runx1* (runt related transcription factor 1) (Fig. 6f and Supplementary Fig. 9c)[47,48]. Additionally, we identified other candidate genes with a potential role in regulating HSC lineage differentiation (Supplementary Data 2). For example, we found *Bcr* (BCR activator of RhoGEF and GTPase) and *Abl1* (c-abl oncogene 1, non-receptor tyrosine kinase), which are necessary for normal neutrophil and lymphoid cell function, respectively (Fig. 6f; Supplementary Fig. 9c). Intriguingly, chromosomal translocation of t(9;22)(q34;q11) results in the fusion oncogene BCR-ABL1, which is characteristic of the HSC-derived chronic myelogenous leukemia, but also present in acute lymphoblastic leukemia[49–51]. Furthermore, we identified *Tet1* (tet methylcytosine dioxygenase 1), suggested as a negative regulator of HSC self-renewal and B cell differentiation, and with a tumor suppressor role in B cell lymphoma[52,53]. Unexpectedly, we detected *Kcnn1* (potassium intermediate/small conductance calcium-activated channel, subfamily N, member 1), which is mainly expressed in the brain but also associated with the bone tumor Ewing sarcoma[54,55]. Its role in hematopoiesis is not well-characterized.

To identify candidate TFs influencing HSC lineage bias and differentiation, we performed motif enrichment analysis of cluster 1 and 2 Lin DARs (Fig. 6g). Intriguingly, the enriched TFBS belonged to the same TF families as the aging related TFs, including ETS-, bZIP-, IRF-, and RUNT-families. Notably, SPI1 (PU.1) and SPIB TFBS were highly enriched in cluster 1, whereas bZIP-, IRF-, and RUNT TFBS were most significantly enriched in cluster 2.

Altogether, our results suggest that CD49b subsets in juvenile mice are epigenetically similar but become more distinct with age. Furthermore, our findings indicate that aging and lineage bias may largely be governed by the same TFs.

## Discussion

As the global population lives longer, health implications due to aging, including cancer, neurodegenerative and chronic diseases, have become a public health concern[4]. Physiologic aging of the hematopoietic system is associated with perturbed immunity and impaired homeostasis, leading to increased risk of blood malignancies, and is attributed to the age-dependent impairment of HSC function[1–3,18]. Notably, there is a predominance of myeloid malignancies with age, whereas the incidence of lymphoid malignancies is higher in children and young people[2,5]. Insights into age-related HSC behavior are critical to understand and overcome physiological consequences of an aging hematopoietic system.

The myeloid predominance in aging was proposed to be due to the reduced ability of HSCs to produce lymphoid cells. However, the discovery of diverse HSC subsets led to a new model proposing that changes in HSC clonal composition cause myeloid predominance[1,2]. Given that many studies investigating the molecular mechanisms of aging have assessed the composite HSC compartment containing diverse HSC subtypes, age-associated features of highly enriched HSCs with distinct behavior are incompletely elucidated. Furthermore, most aging studies using mouse models compare young adult (2–4 months) and old (1.5–2 years) age groups. However, HSCs switch from a fetal to an adult HSC phenotype around 1 month after birth[25]. In juvenile mice, this period involves active tissue growth and is characterized by high self-renewal activity[56], which could contribute to the higher likelihood of lymphoid malignancies in children[5]. This highlights the importance of encompassing the juvenile period when investigating age-associated changes in HSCs.

Here, we have isolated CD49b$^-$ and CD49b$^+$ subsets[23] from the HSC compartment and investigated the cellular and molecular changes across different ages (Fig. 7). Our studies extend on previous findings showing intrinsic functional defects in aged HSCs[7,20] by studying how the highly enriched HSC subsets epigenetically change from juvenile to aged mice. The functional differences between CD49b$^-$ and CD49b$^+$ cells are independent from their spatial localization since no differences were observed in the distribution of CD49b subsets in the BM, contrasting previous reports[57]. The discrepancy is likely due to differences in purity of the HSCs studied. However, the differential adhesion and migration properties of CD49b$^-$ and CD49b$^+$ cells in aging warrant further studies to investigate the role of CD49b in HSC function.

Our studies highlighted a myeloid shift with age, which occurred in both CD49b$^-$ myeloid-biased and CD49b$^+$ lymphoid-biased enriched HSC subsets. Consequently, the lineage distribution was shifted to a more pronounced dominance of M-bi repopulation pattern in CD49b$^-$ HSCs and a switch from mainly a L-bi to a Bal pattern in CD49b$^+$ cells with age. Our data suggest that the CD49b$^-$ subset enriches for M-bi HSCs in all age groups, while the CD49b$^+$ subset enriches for L-bi HSCs in juvenile and adult mice, but marks Bal HSCs in old mice. These

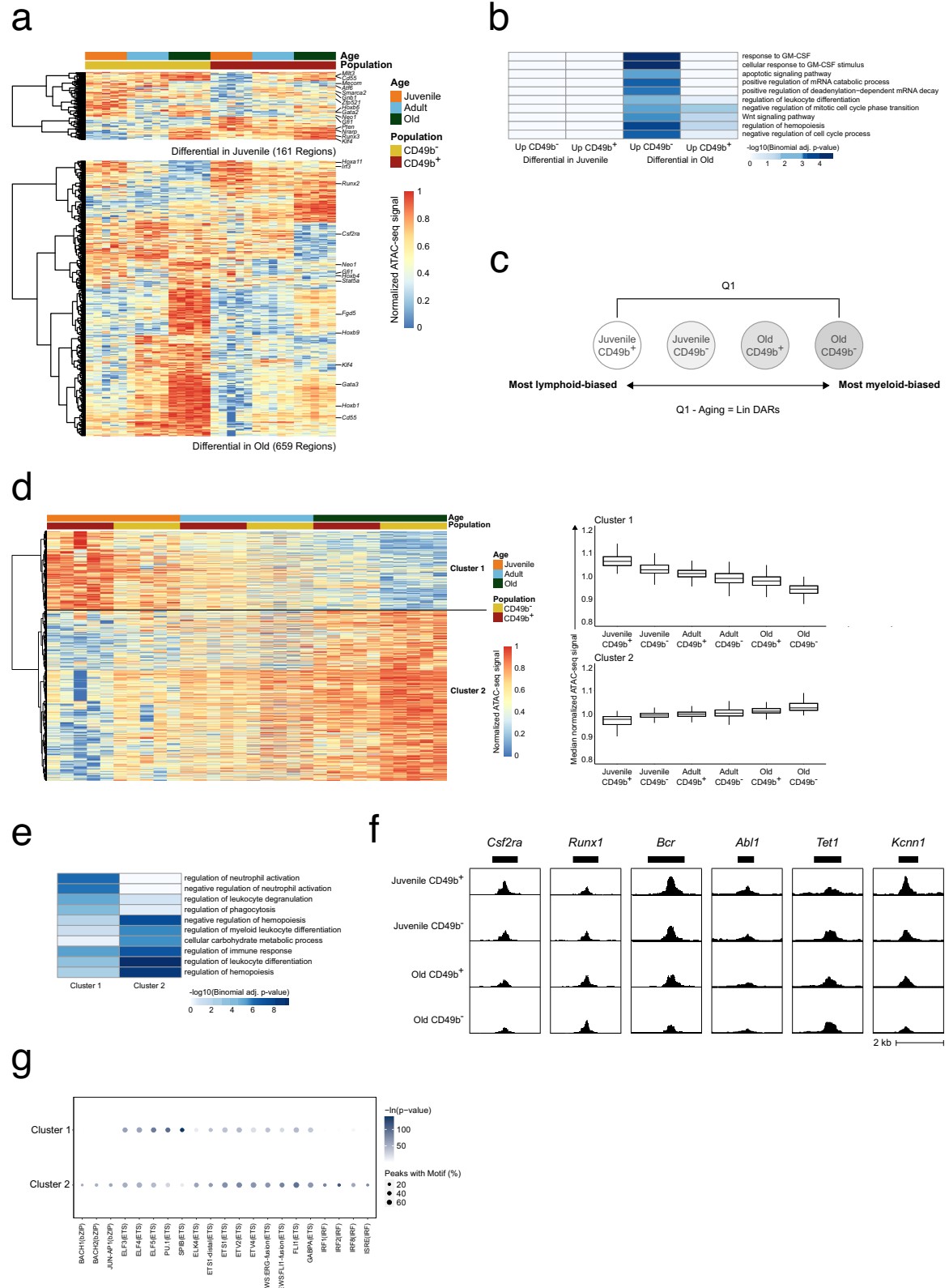

results support that age-related myeloid dominance in the hemato-poietic system stems from intrinsic alterations in lineage differentia-tion properties of HSCs, although changes in the HSC composition cannot be excluded.

We showed that HSCs become more quiescent and less pro-liferative with age, which we substantiated with molecular data. The cell cycling status of aged HSCs has been disputed[8,21,58]. Nonetheless,

our findings align with a recent study suggesting that juvenile mice primarily undergo symmetric self-renewal divisions, while aged HSCs undergo symmetric division to generate two progenitors[56]. This implies that the HSC population gradually expands with age through self-renewal divisions. In aged HSCs, the increased generation of pro-genitors align with loss of stemness and impaired HSC function[1–3,8]. Considering the age-related functional decline but expanded

**Fig. 6 | Aging and lineage bias are regulated by the same transcription factor families. a** Heatmap of row normalized chromatin accessibility for regions with differential accessibility ($p_{adj} < 0.05$, Wald test) between juvenile CD49b$^-$ and CD49b$^+$ HSCs (top), or between old CD49b$^-$ and CD49b$^+$ HSCs (bottom). **b** Top 10 GO biological processes significantly enriched in regions with differential accessibility between CD49b subpopulations in juvenile or old mice. **c** Schematic illustration of the analysis strategy to identify chromatin accessibility changes associated with lineage bias differences (Lin DARs). **d** Heatmap (left) of row normalized chromatin accessibility for Lin DARs. Regions are divided into two clusters based on hierarchical clustering. Boxplots (right) show the median normalized chromatin accessibility in clusters 1 and 2. **e** Top 5 GO biological processes

significantly enriched in clusters 1 and 2. **f** UCSC browser tracks of median ATAC-seq signal for selected Lin DARs. Gene names above the tracks indicate the closest gene to the displayed region. **g** Transcription factors with enriched binding motifs (-ln(p-value)>50) in clusters 1 and 2. A one-sided binomial test was used to determine significance in **b**, **e**, and **g**. p-values in **a**, **b**, **d**, and **e** were adjusted using the Benjamini–Hochberg method. Boxplots show the distribution in each population (center line, median; box limits, interquartile range; whiskers, furthest data point within 1.5× of the interquartile range). Lin DARs, lineage bias associated differentially accessible regions. See also Supplementary Fig. 9. Source data are provided as a Source Data file.

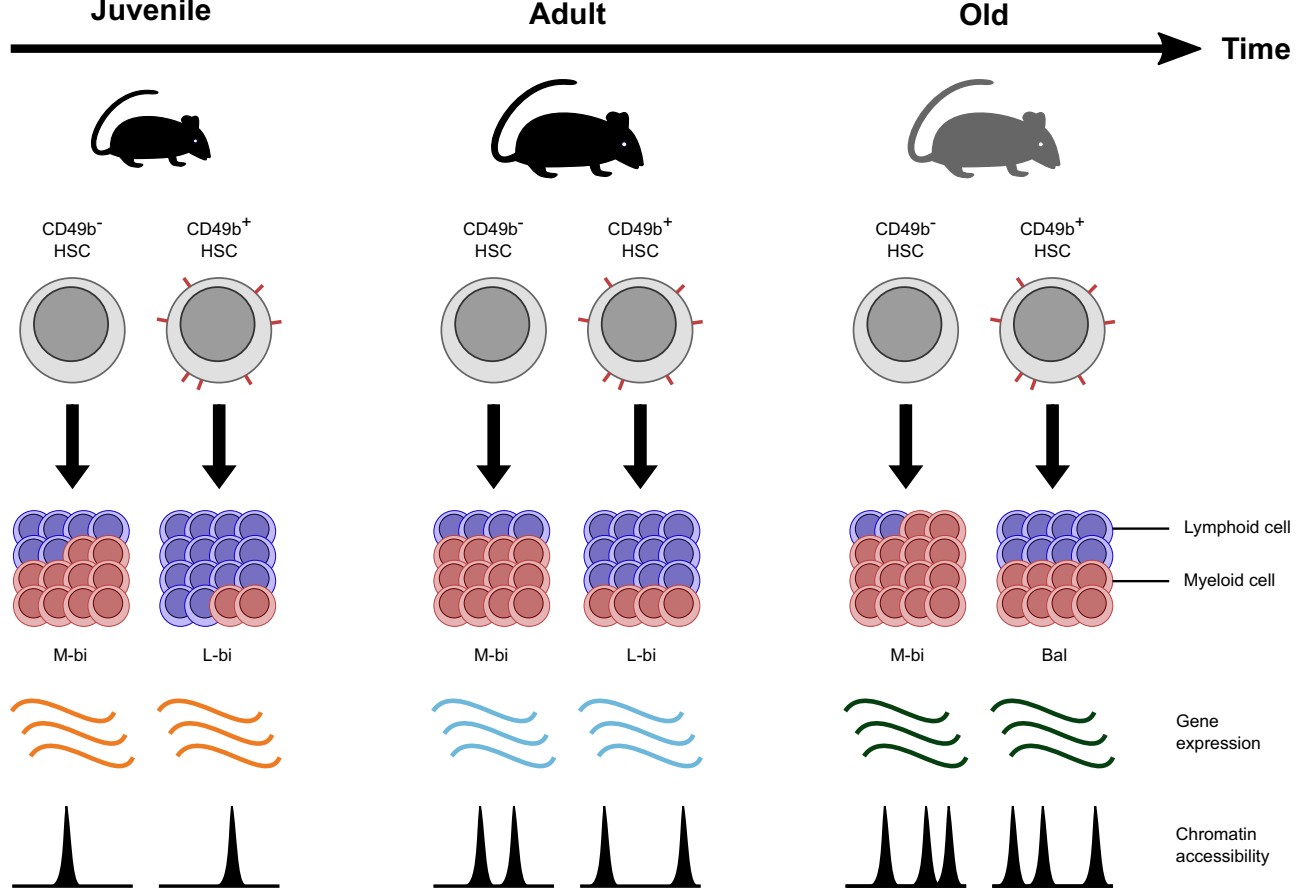

**Fig. 7 | Aging is associated with functional and molecular changes in distinct hematopoietic stem cell subsets.** Schematic overview of the age-related functional and molecular changes in CD49b$^-$ and CD49b$^+$ HSCs in juvenile, adult, and old mice. Aging is associated with gradually increasing myeloid cellular output in

both CD49b$^-$ and CD49b$^+$ HSCs. Gene expression profiling detects age-related molecular changes in total HSCs. Chromatin accessibility analysis reveals age-dependent and CD49b subset-specific differences. M-bi myeloid-biased, L-bi lymphoid-biased, Bal balanced.

phenotypic HSC population, it is conceivable that old HSCs sustain their large population size by limiting cell cycle activity. Interestingly, despite high quiescence in both CD49b subsets, CD49b$^+$ HSCs were less quiescent and more proliferative, while having lower engraftment potential and a lower HSC-score than CD49b$^-$ HSCs. Whether CD49b$^-$ and CD49b$^+$ HSCs differ in their propensity for symmetric or asymmetric division throughout aging, remains to be determined. However, it represents a compelling avenue to explore, which could potentially reveal important mechanisms underlying HSC heterogeneity and in aging.

Lineage bias is a heritable trait[28], indicating that epigenetic mechanisms may regulate HSC heterogeneity[59]. We have previously shown that functionally different HSCs exhibit similar gene expression patterns but distinct epigenetic profiles. Our scRNA-seq analysis captured clear age-related gene expression changes. We also confirmed

that juvenile HSCs are molecularly distinct from fetal HSCs and transcriptionally similar to adult HSCs. Remarkably, we did not detect any global differences between HSC subsets, consistent with the importance of gene regulatory mechanisms governing HSC heterogeneity[23]. Indeed, we observed both age- and subset-related epigenetic differences by ATAC-seq analysis. The chromatin accessibility progressively increased in both HSC subsets, which correlated with the loss of SPIB and SPI1 (PU.1) TFBS. This finding is consistent with the published age-related reduction in PU.1 expression[20,22,60]. Interestingly, we found that reduced SPIB TFBS were already detected in adult HSCs, indicating that age-related remodeling of chromatin regions is initiated in the juvenile-to-adult transition. Our data agree with the hypothesis that some gene expression changes in aging may originate from the juvenile period as part of the growth restricting process that occurs into adulthood[61]. Our results also implicate that aging-induced chromatin

changes could be used to prospectively identify aging epigenetic signatures. Remarkably, age-related molecular changes were primarily observed in HSCs and not propagated to progenitors, suggesting that aging mechanisms preferentially target HSCs. This also implies that it could be sufficient to target HSCs to restore age-related hematopoietic dysfunction. Further functional investigations are necessary to establish whether interference of chromatin remodeling in these regions could be exploited to reverse or remedy aging alterations in HSCs.

Although our ATAC-seq data showed that age-dependent changes were more substantial than changes between CD49b⁻ and CD49b⁺ HSCs, subset-specific epigenetic differences were still detected, which became more distinct in aged mice. Strikingly, the same TF families were enriched in both age-related and subset-specific chromatin regions. Indeed, the ETS family members SPI1 (PU.1) and SPIB, identified to be downregulated with age in our studies, are important for myeloid and lymphoid development and differentiation[62,63]. Notably, although high PU.1 levels are associated with myeloid commitment[64], loss of PU.1 in vivo leads to increased myelopoiesis, impaired lymphopoiesis, and induces myeloid leukemia[65,66], which is compatible with the aging cellular phenotype. These findings highlight the possibility that aging and lineage differentiation processes may involve the same TFs. Our ATAC-seq analysis also identified several candidate genes with possible roles in regulating HSC lineage bias. Among these, *Bcr*, *Abl1*, and *Tet1* are necessary for proper myeloid and lymphoid lineage differentiation, whereas *Kcnn1* has not been studied in the hematopoietic system. Deficiencies or aberrant activation of *Bcr*, *Abl1*, and *Tet1* cause hematopoietic malignancies, highlighting the importance of elucidating the regulatory mechanisms of normal blood lineage differentiation[49–55]. Further functional studies are needed to clarify the involvement of the identified candidates in HSC lineage bias regulation.

Our data suggest that HSC subsets with the same immunophenotype gradually change their functional and epigenetic attributes during aging, starting already in the juvenile-to-adult transition. The intrinsic changes, including alterations in lineage preference, quiescence, and proliferation states, are correlated with significant remodeling of the chromatin landscape. Our findings are therefore compatible with aging models suggesting cell-autonomous HSC changes[2,19]. Collectively, we have demonstrated that CD49b resolves HSC subsets with age-dependent cellular and epigenetic changes (Fig. 7). Our studies provide important insights into the contribution of distinct HSC subsets and the consequences of their functional alterations to the changing hematopoietic compartment in aging. Clarifying the role of HSC subsets in aging is critical towards understanding and providing therapeutic prospects to overcome age-associated dysfunction of the hematopoietic system, including malignancies.

## Methods

### Animals
Female and male C57BL/6J mice used in the experiments were housed and maintained at the Karolinska University Hospital Preclinical Laboratory, Sweden, with the following housing conditions: 12/12 h dark/light cycle, 20 °C ambient temperature, 50 ± 5% humidity. Juvenile mice around 1 month old, adult mice between 2–4 months old, and old mice around 1.5–2 years were used. C57BL/6J or Gata-1 eGFP mice[27], backcrossed >8 generations to C57BL/6J (CD45.2), were used as donors and B6.SJL-*Ptprc^aPepc^b*/BoyJ mice (CD45.1) were used as primary and secondary recipients in transplantation experiments. All experiments were approved by the regional ethical committee, Linköping ethical committee in Sweden (ethical numbers: 882 and 02250-2022).

### Transplantation experiments
In primary transplantation experiments, donor cells (CD45.2⁺) were sorted and intravenously injected with 200,000 CD45.1⁺ support BM cells into irradiated adult CD45.1 mice. Five sorted HSCs from juvenile and adult donor mice and 100 HSCs from old donor mice were transplanted. The full irradiation dose was given in two doses with at least 4 h in between. A total of 10 Gy was used in experiments with juvenile and old donors and total 12 Gy in experiments with adult donors.

In secondary transplantation experiments, 10 ×10⁶ unfractionated BM cells, from phenotypic HSC (LSK CD48⁻CD150⁺) reconstituted primary recipients, were intravenously injected into 1–5 lethally irradiated secondary recipients. Recipient mice were monitored and PB analyses were done regularly up to 5–6 months post-transplantation for both primary and secondary transplantation experiments.

### Preparation of hematopoietic cells
Bone marrow (BM) single cell suspensions were prepared by crushing femurs, tibiae, and iliac crests isolated from the mice into Phosphate-Buffered Saline (PBS, Gibco) supplemented with 5% Fetal Bovine Serum (Gibco) and 2 mM Ethylenediaminetetraacetic acid (EDTA, Merck) (PBS/5%FCS).

Unfractionated BM cells were counted on the XP-300-Hematology Analyzer (Sysmex Corporation) and then Fc-blocked either with purified CD16/32 (BD Biosciences) or stained with CD16/32 antibody conjugated to a fluorophore. Following Fc-block, the cells were stained with antibodies against cell surface marker antigens. See Supplementary Data 3 for antibodies used and Supplementary Table 1 for phenotypic definitions of hematopoietic populations.

To detect HSCs, unfractionated BM cells were enriched with CD117 MicroBeads (Miltenyi Biotec) and subsequently selected using immunomagnetic separation before staining with antibodies against cell surface markers.

Peripheral blood (PB) was sampled from the tail vein of transplanted mice. Blood was collected in lithium heparin coated microvette tubes (Sarstedt). The platelet fraction was separated from whole blood samples by centrifugation, and leukocytes were subsequently isolated with Dextran sedimentation. Isolated platelets, erythrocytes, and leukocytes were stained with antibodies against cell surface antigens as previously described[23].

### Flow cytometry analysis of hematopoietic cells
Flow cytometry analyses were done on FACSymphony™ A5 and LSR Fortessa™ following cell preparation. For cell sorting experiments, FACSAria™ Fusion cell sorters (BD Biosciences) were used. The mean cell sorting purity was 93% ± 6%, calculated from 34 experiments. Fluorescence minus one (FMO) controls were included in every experiment. For single cell in vitro experiments, single cell sorting into individual wells of 96-well or 72-well plates was confirmed by sorting 488-nm fluorescent beads (ThermoFisher Scientific). For Smartseq2 single cell sorting into 384 wells, a colorimetric test was performed to validate sorting efficiency. Analyses following data acquisition were done using FlowJo software version 10 (BD Biosciences).

### Calculation of reconstitution and lineage bias
Donor reconstitution was calculated based on the frequency of CD45.2⁺ events in total leukocytes. Transplanted mice with ≥0.1% total donor contribution in leukocytes (CD45.2⁺) and/or platelets in the peripheral blood (PB), represented by ≥10 events in the donor gate of the recipient mice, were scored as positively repopulated at month 2 post-transplantation.

Blood lineage repopulation was calculated based on the frequency of CD45.2⁺ events in the leukocyte lineages or of Gata-1 eGFP⁺ events in platelets and erythrocytes. Transplanted mice were determined to be positive for a specific blood lineage when the repopulation was ≥0.01% and represented by ≥10 events in the donor gate.

Relative donor reconstitution levels were calculated based on the frequency of B, T, NK, and myeloid cells within the CD45.2⁺ cells at month 5–6 post-transplantation.

The blood lineage distribution of the leukocyte fraction was calculated based on the ratio of lymphoid (L) to myeloid (M) cells (L/M) in the PB of adult or old unmanipulated mice. Lymphoid cells included B, T, and NK cells. The calculated L/M ratios from unmanipulated mice (Supplementary Fig. 3e) were used to categorize the lineage distribution in the PB of transplanted mice 5–6 months post-transplantation.

## Analysis of bone compartments

To analyze the distribution of CD49b[−] and CD49b[+] cells in different bone compartments, femurs from mice of each age group were isolated. The trabecular bones were cut off from both ends of the femur using scissors. The central marrow was then flushed out with PBS/5% FCS and resuspended in single cells using a 23–25 G needle and syringe (Supplementary Fig. 5a). The flushed bones and trabecular bones were then crushed in PBS/5%FCS to isolate hematopoietic cells from the endosteal and trabecular bone compartments, respectively. Single cell suspensions from the central marrow, endosteum, and trabecular bone compartments were then counted and Fc-blocked. Subsequently, the cells were stained with antibodies against HSC cell surface marker antigens (Supplementary Data 3) and analyzed by flow cytometry.

## In vitro assays

Myeloid (CD11b[+]Gr-1[+] and/or F4/80[+]CD11b[+]) and B cell (CD19[+]B220[+]) differentiation potential was assessed using the OP9 co-culture assay by sorting single cells onto OP9 stroma and analyzed after 3 weeks of culture[23]. Megakaryocyte potential was assessed by single cell sorting or manual seeding at 1 cell per well into 72-well plates (ThermoFisher Scientific) and evaluated after 11 days by scoring the presence of megakaryocytes using an inverted microscope[23]. Comparable results were obtained with manual plating when corrected for the Poisson distribution, as previously described[67]. Cell division kinetics were carried out by tracking cell divisions of single plated cells in 60-well plates (ThermoFisher Scientific) for 3 days post-sort. See Supplementary Table 2 for culture conditions.

## Cobblestone-Area-Forming Cell (CAFC) assay

Transmigration of CD49b[−] and CD49b[+] HSCs was evaluated by a modified CAFC assay using primary BM MSCs, as previously described[68]. Mouse BM MSCs[68] were sorted (Supplementary Data 3 and Table 1) and seeded at 5000 cells/well in complete DMEM in a 96-well plate (167008; Thermo Scientific) and culture media was removed after two days. Subsequently, 100 FACS-sorted CD49b[−] or CD49b[+] HSCs were seeded and co-cultured with the MSCs in 100 uL/well of complete Myelocult medium. Non-migrated cells were counted 24 h and 48 h after co-culture to assess cell adhesion and migration. The migrated HSCs with CAFC capacity were extrapolated at 28 days post co-culture based on the number of CAFCs formed. The CAFCs were counted under an inverted microscope using phase contrast. A cell cluster containing more than three cobblestone-like cells beneath the MSCs was defined as a CAFC. See Supplementary Table 2 for culture conditions.

## Cell cycle and proliferation assays

The cell cycle state of HSCs was analyzed using Ki-67 staining, following the manufacturer's protocol from the BD Cytofix/Cytoperm Kit (BD Biosciences)[23]. The cell proliferative state of HSCs was assessed using 5-Bromo-2'-deoxyuridine (BrdU) incorporation, where one dose of BrdU was given by intraperitoneal injection (50 mg/g bodyweight, BD Biosciences), followed by oral administration via drinking water (800 mg/mL, Merck) for 3 days post-injection. The BrdU experimental process and visualization were performed according to the BrdU Flow Kit protocol (BD Biosciences).

## Single cell RNA- and ATAC-sequencing

Single cells were deposited into 384-well plates for SmartSeq2 single cell RNA-seq as previously described[23]. Libraries were sequenced on HiSeq3000 (Illumina) using dual indexing and single 50 base-pair reads. See Supplementary Data 1 for sequenced RNA-seq samples.

Reads were demultiplexed, aligned to the mm10 reference genome using TopHat (v2.1.1), and deduplicated using SAMtools (v0.1.18). For further analysis, R and the Seurat package (v.4.3.0) were used. Reads mapping to *CT010467.1* were excluded as they largely originate from rRNA contamination. Cells were filtered to have 50,000–750,000 reads, <10% mitochondrial reads, and <10% ERCC spike-in contribution. Lowly expressed genes with ≤400 reads across all cells were filtered out. Data were normalized using Seurat's SCTransform, regressing out the percentage of mitochondrial reads. PCA was run on the 3000 most variable genes and the top 10 PCs were used as input for UMAP plots. Differential expression analysis was done with Seurat's FindMarkers function using a logistic regression framework (LR test) the and the percentage of mitochondrial reads as latent variable. Gene set enrichment analysis (GSEA) was run using the GSEA desktop application (v4.3.2). Module scores for indicated gene sets were calculated using Seurat´s AddModuleScore function. To annotate cells based on similarity in expression compared to published scRNA-seq data the SingleR package (v2.0.0) was used with de.method set to"wilcox". Previously published scRNA-seq from adult and old HSCs was analyzed from counts tables by filtering cells as described in the individual studies and integrating all data using Seurat´s anchor-based integration workflow (using the FindIntegrationAnchors and IntegrateData functions). HSC-scores were calculated using the hscScore tool[41] using the provided Jupyter notebook and trained model, as suggested by the developers.

Bulk ATAC-seq was performed with 500 sorted cells using a modified Omni-ATAC protocol, as previously described[23]. Samples were paired-end sequenced (2 × 41 cycles) on NextSeq 500 (Illumina). See Supplementary Data 2 for sequenced ATAC-seq samples.

Sequence reads were aligned to the mm10 reference genome and peaks were called using the nf-core ATAC-seq pipeline (v1.0.0). Poor quality samples (Fraction of Reads in Peaks, FRiP, <10%) were excluded after quality check. Read coverage was normalized to 10⁶ mapped reads in peaks, and median normalized read coverage was calculated and visualized using the UCSC genome browser (https://genome.ucsc.edu/). Read positions were adjusted by +4 bp and −5 bp for the positive and negative strand, respectively. HOMER (v4.11) was used to annotate peaks and quantify reads in consensus peaks. Peaks with more than 5 fragments per kilobase million (FPKM) in at least a third of the samples were considered found in a population. Peaks not found in any population were excluded. R was subsequently used to log₁₀ transform and quantile normalize read counts for visualization. Differentially accessible regions were determined using the DESeq2 package (v1.38.2). Differential regions not found in either compared population were excluded from further analysis. Adult ATAC-seq data from Somuncular et al.[23] were included in the analysis. To match the sample size and quality from the juvenile and old age groups, only 5 adult samples with the highest FRiP scores for each cell population were included. GREAT (http://great.stanford.edu/public/html/, v4.0.4) was used for gene ontology analysis using online default settings. Motif enrichment analysis was done using HOMER. ATAC-seq data from Itokawa et al.[22] was reprocessed from fastq files as described above and integrated by quantifying reads in a combined set of consensus peaks. Batch effects between the studies were removed using the limma package (v3.54.2), and data was collectively visualized.

## Statistical analysis

GraphPad Prism v.9.4.1 for Mac OS or R was used for all statistical analyses. Measurements were taken from distinct samples except

for the cell division and transplantation data, where the same cell/mouse was measured repeatedly. The Shapiro–Wilk test was used to test for normal distribution. Parametric tests were performed with one- or two-way ANOVA, t-tests, or the Wilcoxon signed-rank test if the normality assumption was fulfilled. Non-parametric tests were performed with the Mann–Whitney or the Kruskal–Wallis with Dunn's post-hoc test. For binary data, Fisher's exact test was used. Where necessary, *p*-values were adjusted for multiple testing. All tests, except motif enrichment and gene ontology analyses, were carried out two-sided. Mean ± SD and *p*-values are indicated in all figures.

### Reporting summary

Further information on research design is available in the Nature Portfolio Reporting Summary linked to this article.

## Data availability

Juvenile and old mice ATAC-seq and scRNA-seq data are deposited in the European Nucleotide Archive (ENA) under the accession number PRJEB55627. Adult ATAC-seq and scRNA-seq data have been previously deposited in ENA under the accession number PRJEB47791[23]. Source data are provided with this paper.

## Code availability

The code used for analysis is available on GitHub at: https://github.com/KI-LucLab/Aging_is_associated_with_functional_and_molecular_changes_in_distinct_HSC_subsets. The input data can be found at: https://doi.org/10.5281/zenodo.13378750.

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

## Acknowledgements

The authors thank Claus Nerlov (University of Oxford) for providing the Gata-1 eGFP mouse strain; Joakim Dillner (Karolinska Institutet, KI) for sequencing support; Preclinical Laboratory, Karolinska University Hospital and KI MedH Flow Cytometry Core Facility for their services; Laura Covill (KI) and Hui Gao (KI Centre for Bioinformatics and Biostatistics (CBB)) for helpful statistical advice; KI Single cell core Facility @ Flemingsberg campus (SICOF) for their single cell sequencing services. The computations and data storage were enabled by resources provided by the National Academic Infrastructure for Supercomputing in Sweden (NAISS) and the Swedish National Infrastructure for Computing (SNIC) at Uppmax partially funded by the Swedish Research Council (2022-06725 and 2018-05973). S.L. was supported by a Wallenberg Academy Fellow award (2016.0131) and by the Swedish Childhood Cancer Fund (TJ2017-0074, PR2017-0047). J.H., E.L. and T.-Y.S were supported by KI Doctoral Education grants. This work was supported by grants from the European Hematology Association (S.L.); the Swedish Cancer Society, CAN2017/583 (S.L.), 20 1062 PjF (S.L.), 19 0092 SIA (H.Q.), 20 1222PjF (H.Q.); the Swedish Research Council, 2016-02331 (S.L.), 2022-01228 (H.Q.); the Strategic research area (SFO) in Stem cell and Regenerative Medicine (S.L.); Åke Olsson foundation (S.L.); Åke Wiberg foundation (S.L.) and King Gustav V Jubilee Fund (R.M.).

## Author contributions

Contribution: E.S., T.-Y.S., Ö.D., E.L., A.-S.J. and S.L. performed experiments and analyzed data; C.G., E.T. and A.F. assisted in experiments; J.H., and T.-Y.S. performed bioinformatics analysis; A.-S.J., H.Q., R.M. and S.L. supervised the work; S.L. designed the project; S.L. wrote the paper together with E.S., J.H., T.-Y.S., A.-S.J., H.Q. and R.M.; all authors reviewed the manuscript before submission.

## Funding

## Competing interests

The authors declare no competing interests.
