## [Peer Review File · Nature Communications]

REVIEWER COMMENTS

Reviewer #1 (Remarks to the Author):

The manuscript 'Aging is associated with functional and molecular changes in distinct hematopoietic stem cell subsets' by Somuncular et al. investigates by how different subtypes of murine HSCs, which are separated by CD49b, change upon aging. The manuscript is organized in a logical flow and the experimental set-ups are sound. However, within the data and thus also within the manuscript, no new principles or techniques are reported and there are no mechanistical studies. The data do not really provide new approaches for intervention as claimed in the discussion. It remains descriptive work, and the results reflect/confirm well established data of HSC aging, while all the young HSC information has been previously published by the group (PMID 35714596). The general observation that aging affects both lymphoid and myeloid biased HSCs at a similar level was published by Dykstra et al. (PMID 22110168). The authors cite that publication, but never discuss it in detail. There are many expression panels published on murine young and aged HSCs, also nicely summarized by PMID 33876187, which are not really discussed either.

Additional comments:

Marker selection: It is not clear whether CD49b can be regarded as a reliable lymphoid marker if the association is lost upon aging. The loss of CD49b as a differentiation marker turns the MS into a young vs. aged HSC analysis, as the authors simply combine the data for further analysis. Similarly, CD41 (myeloid marker) illustrates changes upon aging. How does CD41 expression (myeloid marker) correlate to the CD49b expression in young?

This reviewer does not understand at all why for young and adult only 5 HSCs were transplanted and for old 100? The transplantation results can thus not be compared, as we go from a likely monoclonal/low oligoclonal reconstitution (5 cells) to a multiclonal reconstitution (100 cells). For the transplantation experiments summary graphs might be helpful to provide the reader with a better understanding of the overall difference, not just bits and pieces.

What is the reason to provide additional 3 days of BrdU in drinking water after an initial i.p. injection? How can such data be interpreted?

In some of the figures the adult samples are missing, even so the data is already published it is good to observe the dynamics within one graph. Also, to better define what are differences already arising from young to adult, and what are differences from adult to old.

Several experiments show only n=2 for repeats, a higher number of repeats, especially in the heterogenous world of aging, might turn the data more solid.

The difference in Progenitor/Lin DARs vs HSCs (line 261-263) provide a really interesting novel view, and could indeed form that starting point for a novel project, if pursued in detail.

Reviewer #2 (Remarks to the Author):

The study performed by Somuncular et al.; focuses on addressing an open and relevant question in the Hematopoietic Stem Cell (HSC) aging field, whether upon aging myeloid-skewing arises from life-long expanding platelet/myeloid-biased HSC clones or if aged HSCs strive to differentiate to lymphoid progenitors at a population level.

The authors reflect on the fact that epigenetic differences can be a determinant factor, however efforts to separate distinct epigenetically HSC populations has not been yet accomplished. Following up on a previously published study (10.1016/j.stemcr.2022.05.014), the authors embark on a comprehensive study on the molecular and functional characterization of HSCs separated by CD49b – encoded by the *Itgb2* gene – now looking at age differences between young, adult and aged HSCs. This study demonstrates that although CD49b- and CD49b+ HSCs have a similar transcriptome, there are significant epigenetic differences between these populations, and these are deepened upon aging. Functionally, CD49b- HSC demonstrate more “aging” traits such as myeloid-bias, early exhaustion upon serial transplantation.

Although the present study provides further relevant clues about the mechanisms whereby myeloid skewing arises, it misses on the opportunity to explore novel and different angles which could have shed further evidence of the relevance of CD49b's function within the HSC compartment.

The present study has a major focus on epigenetic characterization, but the evidence that there are no major transcriptional changes between CD49b populations weakens the grounds for this pursue. Since the protagonist of the study is CD49b – a integrin – it is surprising that the authors do not further investigate the actual function in terms of adhesion/signaling properties. A reasonable hypothesis could be that CD49b populations have different integrin subunit composition in their membrane which in turn would yield different adhesive properties, potentially having a profound impact on the spatial localization of these cells in the BM and therefore on the HSC behaviors (doi.org/10.1016/j.stem.2013.06.015).

Although the current manuscript has an impressive body of work, characterization of these population in the BM niche context would significantly strengthen the argument that CD49b is a key role player during the aging process, analogously to similar studies (10.1016/j.stem.2010.11.030 and more recently 10.1038/s41467-020-20801-0).

Taken together, I would consider this study for publication at Nat. Comms. with substantial revisions. Specifically, further studies focusing on the role of CD49b in terms of adhesion (e.g. in vivo or in vitro homing studies, BM niche localization of CD49b⁻ vs. CD49b⁺ populations).

Beyond the aforementioned points:

Conceptual Comments:

1. It seems that the ratio between CD49b⁻ and CD49b⁺ is of essence. Thus, showing these groups separated by age is distracting, I would suggest replotting the data grouping based on CD49b expression rather than by age. Comparisons would be clearer. For example, Figure 1F seems to demonstrate that in old mice, CD49b⁺ incorporate more BrdU than CD49b⁻, but this is not clear.
2. In Figure 1, the authors assess frequencies and cells cycle stages of CD49b⁻ and CD49b⁺ within the LSK/SLAMhigh/CD34⁻ gate. However, it has been reported by different research groups ([10.1182/blood-2008-12-192054](https://doi.org/10.1182/blood-2008-12-192054) and doi.org/10.1073/pnas.1000834107) that SLAMhigh cells are myeloid-bias. One would predict that with the current gating strategy CD49⁺ cells which are not myeloid-bias and potentially CD150^{low}, yet still hold stem cell potential are being excluded for the analyses. Would it be any difference in terms of frequency or ration between CD49b⁻ and CD49b⁺?
3. Still in Figure 1, it is demonstrated that there is a significant expansion in absolute number of both CD49b subsets upon aging, yet the authors conclude that the same populations become more quiescent, which seems a contradictory. How the authors explain these findings? Moreover, the authors should also contextualize their cell cycle findings in light with conflicting previous publications where authors demonstrate that HSC become more or less cycling upon aging.
4. In Figure 4 the authors performed scRNA-seq in different progenitor populations. Due to HSC pool heterogeneity, specially upon the aging process, it would be interesting to demonstrate how the author's data stand compared to different already published data sets (e.g. [10.1186/s12915-021-00955-z](https://doi.org/10.1186/s12915-021-00955-z)). The authors can also explore different previous findings ([10.1038/nature25168](https://doi.org/10.1038/nature25168)) to further corroborate with their cell cycle findings. By comparing own data to previously published datasets would not only strengthen the author's argument but also would help to contextualize these findings.
5. For the ATAC-seq data which is described in Figures 5 and 6, how does the data correlates with previously published ATAC-seq from young and aged mice? doi.org/10.1038/s41467-022-30440-2

Minor comments:

6. In Line 84 replace “development” as this seems to indicate that the study is about the early life events, not as in lifespan.

7. The references 12 and 23 do not mention CD49b in line 77, please reframe it.

8. In Figure 2C, the chimerism plot should not be log-transformed as it's hard to see potential differences.

Reviewer #3 (Remarks to the Author):

The authors Somuncular, Hauenstein, Su et al. describe the functionality of two distinct HSC subtypes (CD49b[±]) during aging and how aging alters their lineage commitment. The authors clearly state transcriptional similarities but emphasize differences on the epigenetic level between young and old HSC subtypes. Their results demonstrate how myeloid lineage bias could be affected by lineage specific transcription factors and higher accessibilities for their corresponding binding motifs with age.

The presented data is of high quality, obtained with appropriate techniques and supports the authors' conclusions. Some of the provided figures are missing statistical analysis, which will be commented on in the section below. Since the impact of aging on HSC functionality and diversity is broadly discussed and has a wide impact on the field of hematopoietic aging, the community highly profits from the presented results.

Said that, this reviewer hopes that the following comments help to improve the clarity of this manuscript.

1) The authors describe an increase of CD49b[±] HSCs with age (Fig. 1B,C). The overall contribution of CD49b[±] cells to the composition of the aging HSC pool would be easier to judge if the percentage of CD49b[±] cells within the HSC pool would be graphically displayed between young, adult and old mice.

2) While the authors clearly show that the cycling activity of the HSC subtypes changes with age, this reviewer is missing a statement of the authors discussing the high quiescence (Fig. 1D,E,F) but reduced HSC-score (Fig. 4D) of old CD49b⁺ cells regarding their self-renewal potential also considering previously published work describing aged HSCs to be frequently cycling (e.g. Morrison et al., 1996).

3) The provided data for in vitro functionality of young CD49b⁺ HSCs, which have been described to have lymphoid bias, show lower B cell frequencies compared to myeloid frequencies (Fig. 2A). To strengthen the aspect of CD49b⁺ being lymphoid-biased HSC subsets, the manuscript would benefit from a comment on this lack of lymphoid over myeloid differentiation potential for young CD49b⁺ HSCs. In case the provided format of data display is misleading and indeed the lymphoid differentiation is significantly increased, this reviewer would kindly ask the authors to rearrange the data display and to add necessary statistical comparisons.

4) The authors further transplant CD49b[±] cells to follow their reconstitution potential (Fig. 2D) but statistical analysis showing significant changes for the reconstitution of the myeloid vs. lymphoid compartment is missing and should be included to support their findings.

5) The authors nicely show the reconstitution of the myeloid vs HSC compartment by CD49b[±] cells in elegant secondary transplantations (Fig. 3C). However, this reviewer is missing a statistical comparison to judge the outcome of this experiment. As the data is presented, the authors show higher reconstitution of the HSC compartment by old CD49b⁺ HSCs compared to young CD49b⁺ HSCs and equal reconstitution of the myeloid and HSC compartment by CD49b⁻ old and young HSCs. This reviewer is missing a critical discussion with previously published literature describing reduced reconstitution by old HSCs (e.g. Dykstra et al., 2011, Morrison et al., 1996, Poscablo et al., 2021).

6) Statistically comparing the HSC-score (Fig. 4D) between young and old CD49b[±] would improve the message of this figure.

7) The connection of higher chromatin accessibility in old HSC to myeloid differentiation is clearly demonstrated (Fig. 5). The meaning of this outcome should be proven by including absolute numbers of myeloid cells reconstituted by CD49b⁺ vs CD49b⁻ young vs old donor HSCs in the peripheral blood after transplantation.

8) The authors convincingly show a decrease in PU.1 expression in old HSCs (extended Fig. 7). This reviewer is missing a comment on previously published work showing how high PU.1 levels instruct myeloid differentiation (e.g. Nerlov and Graf, 1998) which might be counterintuitive regarding the described myeloid bias of old HSCs in this manuscript. This is especially interesting, as the authors present the highest motif enrichment for PU.1 binding sites in young lymphoid biased CD49b⁺ cells (Fig. 6G).

9) As a minor comment for extended figure 3C, this reviewer is missing the legend explaining the color code.

This reviewer highly recommends including missing statistical comparisons as well as a critical discussion about possible controversies between previously published work and new data provided in this manuscript to strengthen the important outcome of this study.

Dear Editor and Reviewers,

We would like to kindly thank the reviewers for their thoughtful and insightful feedback on our manuscript entitled “Aging is associated with functional and molecular changes in distinct hematopoietic stem cell subsets”.

Following the suggestions of the reviewers, we have included additional experiments and analyses to address the concerns that were brought up and revised the manuscript accordingly.

Here we provide our point-by-point response to the reviewers’ comments.

Reviewer #1:

The manuscript ‘Aging is associated with functional and molecular changes in distinct hematopoietic stem cell subsets’ by Somuncular et al. investigates by how different subtypes of murine HSCs, which are separated by CD49b, change upon aging. The manuscript is organized in a logical flow and the experimental set-ups are sound. However, within the data and thus also within the manuscript, no new principles or techniques are reported and there are no mechanistical studies. The data do not really provide new approaches for intervention as claimed in the discussion. It remains descriptive work, and the results reflect/confirm well established data of HSC aging, while all the young HSC information has been previously published by the group (PMID 35714596). The general observation that aging affects both lymphoid and myeloid biased HSCs at a similar level was published by Dykstra et al. (PMID 22110168). The authors cite that publication, but never discuss it in detail. There are many expression panels published on murine young and aged HSCs, also nicely summarized by PMID 33876187, which are not really discussed either.

We thank the reviewer for the comments on our manuscript. In our previous publication (Somuncular *et al.*, *Stem Cell Rep.*, 2022) we identified that CD49b subfractionated the HSC compartment into functionally distinct HSC subsets in adult mice. In the current study, we have investigated how these HSC subsets change throughout aging by examining juvenile (1 month), adult (2-4 months), and old mice (1.5-2 years). We acknowledge that the term “young” is typically used to denote young adult mice in most aging studies, which could lead to confusion given our usage of the term to refer to juvenile mice. To improve clarity, we have replaced “young” with “juvenile” throughout the revised manuscript.

The reviewer rightly points out that aging due to intrinsic changes in both lymphoid- and myeloid-biased HSCs have previously been published by Dykstra *et al.*, *J. Exp. Med.*, 2011. However, another widely accepted model describes aging as a result of changes in HSC clonal composition, in which the functional properties of individual HSCs remain unchanged (Geiger *et al.*, *Nat. Immunol.*, 2013). Therefore, given that the mechanism underlying HSC aging has not been established, our study holds significant relevance. Our current study extends upon the findings by Dykstra *et al.* by evaluating significantly enriched HSC subsets and analyzing their molecular regulation in juvenile, adult, and old mice. The novelty of our study lies in the inclusion of highly enriched HSC subsets, the juvenile age group, and the comprehensive molecular analysis of HSCs throughout aging using both scRNA-seq and ATAC-seq analyses. We have clarified this in the Discussion, pages 15-16, lines 367-377.

While many expression studies have been published, as the reviewer notes, including Svendsen *et al.*, *Blood*, 2021, our study demonstrates that gene expression analysis can only identify age-associated changes, and fails to distinguish functionally diverse HSC subsets. In contrast, our study shows that the use of ATAC-seq can detect both age- and subset-specific changes in the HSC compartment. Furthermore, we establish that the inclusion of the juvenile age group is crucial to assess age-related differences in the hematopoietic system, as chromatin accessibility changes are already evident in the juvenile-to-adult transition. Consequently, aging epigenetic signatures can be deciphered early in lifespan, providing unique opportunities to

explore rejuvenation strategies. In the manuscript, we underscore several findings from our analysis that align with existing literature, thereby validating our results. We also highlight new and unexplored targets, which will require functional testing for validation. Such experiments may include knock-out/knock-down, overexpression, and/or inhibitor experiments. However, these approaches would necessitate a significant amount of time, especially considering that serial transplantation experiments would also be required. We believe these experiments exceed the scope of this study, and we hope the reviewer agrees.

Additional comments:

1. Marker selection: It is not clear whether CD49b can be regarded as a reliable lymphoid marker if the association is lost upon aging. The loss of CD49b as a differentiation marker turns the MS into a young vs. aged HSC analysis, as the authors simply combine the data for further analysis. Similarly, CD41 (myeloid marker) illustrates changes upon aging. How does CD41 expression (myeloid marker) correlate to the CD49b expression in young?

In our study, we are showing that lack of CD49b (CD49b⁻) in the HSC compartment reliably marks cells enriched for myeloid bias across juvenile, adult, and old mice (Fig. 2f). In contrast, CD49b⁺ cells enrich for lymphoid-biased HSCs in juvenile and adult mice, but mark lineage-balanced HSCs in old mice (Fig. 2f,g and Supplementary Fig. 3f). CD49b is therefore a useful marker to distinguish functionally diverse HSC subsets across different ages and is thus not exclusively a lymphoid marker. We have described this finding on pages 6-7, lines 136-145 in the Results and on page 16, lines 386-388 in the Discussion.

As suggested by the reviewer, we also analyzed the CD41 expression in juvenile, adult, and old mice. CD41 expression was very low in juvenile mice but increased with age as expected. Notably, the CD41 expression was higher in old CD49b⁻ HSCs compared to old CD49b⁺ HSCs, which is in agreement with CD49b⁻ identifying myeloid-biased cells and CD49b⁺ marking lineage-balanced HSCs in aged mice. We have included the CD41 expression analysis in a new figure (Supplementary Fig. 4e) and described the data on page 7, lines 150-152.

2. This reviewer does not understand at all why for young and adult only 5 HSCs were transplanted and for old 100? The transplantation results can thus not be compared, as we go from a likely monoclonal/low oligoclonal reconstitution (5 cells) to a multiclonal reconstitution (100 cells). For the transplantation experiments summary graphs might be helpful to provide the reader with a better understanding of the overall difference, not just bits and pieces.

Our primary objective was to assess the lineage bias differences between CD49b subsets and compare lineage bias distribution across the age groups. Given the diminished regenerative capacity of old HSCs, we transplanted 20 times more old HSCs. This approach was taken to ensure comparable repopulation level to juvenile and adult counterparts, thereby preventing any skewing of the lineage bias distribution data arising from mice with low engraftment or significant variation in repopulation levels, as would be expected if transplanting low numbers of old HSCs. Indeed, we were able to achieve comparable repopulation levels across all age groups despite the higher number of old HSCs being transplanted, confirming the impaired HSC function in aged mice (Fig. 2c).

In our lineage distribution analysis, we observed a clear increase in myeloid contribution in mice transplanted with old HSCs. To improve data clarity and interpretation, we have incorporated a summary graph into Fig. 2e.

3. What is the reason to provide additional 3 days of BrdU in drinking water after an initial i.p. injection? How can such data be interpreted?

Considering that the CD49b⁻ and CD49b⁺ HSCs are highly quiescent (Fig. 1d) and because not all HSCs enter cell cycle at once (Fig. 1e), we provided the mice with an additional 3 days of BrdU in the drinking water in conjunction with BrdU intraperitoneal injection to increase labeling, as described in Matatall *et al.*, *Methods Mol. Biol.*, 2018. This protocol enabled a sufficient number of labelled HSCs for analysis and quantification.

4. In some of the figures the adult samples are missing, even so the data is already published it is good to observe the dynamics within one graph. Also, to better define what are differences already arising from young to adult, and what are differences from adult to old.

While we did not initially include the adult age group in all experiments due to our detailed characterization of this age group in a previous study (Somuncular *et al.*, *Stem Cell Rep.*, 2022), we acknowledge the value of including it to observe age-related dynamics. Although the main focus of this paper was to compare the juvenile and old mice, we have now incorporated the adult age group from the Somuncular *et al.* study, where possible, to improve clarity and to provide a comprehensive view of the data. The new figures are presented in Fig. 6d, Supplementary Figs. 8b,c,e,9b.

5. Several experiments show only n=2 for repeats, a higher number of repeats, especially in the heterogenous world of aging, might turn the data more solid

We appreciate the reviewer's comment regarding the reliability and reproducibility of our data. In response, we have included additional experiments to ensure at least 3 experimental repeats (refer to Figs. 1b-1f, 2a-2b, Supplementary figs. 1c, 1e, 1g, 3d, 4h, 4i). Unfortunately, due to limitations related to animals of the appropriate age of the Gata-1 eGFP donor mice, we were unable to perform further transplantation experiments for the juvenile and adult age groups within a reasonable revision timeframe. However, we would like to point out that the adult age group has been thoroughly characterized in our previous publication (Somuncular *et al.*, *Stem Cell Rep.*, 2022). Furthermore, we have transplanted at least 14 mice with juvenile CD49b HSC subsets in two separate experiments. The congruence of our molecular results with the juvenile transplantation findings further substantiates the validity of our data.

6. The difference in Progenitor/Lin DARs vs HSCs (line 261-263) provide a really interesting novel view, and could indeed form that starting point for a novel project, if pursued in detail.

We thank the reviewer for the positive response to the differences observed in our progenitor vs. HSCs analyses. We agree that this is indeed an interesting and novel finding and intend to further pursue the findings in our follow-up studies.

Reviewer #2 (Remarks to the Author):

The study performed by Somuncular et al.; focuses on addressing an open and relevant question in the Hematopoietic Stem Cell (HSC) aging field, whether upon aging myeloid-skewing arises from life-long expanding platelet/myeloid-biased HSC clones or if aged HSCs strive to differentiate to lymphoid progenitors at a population level.

The authors reflect on the fact that epigenetic differences can be a determinant factor, however efforts to separate distinct epigenetically HSC populations has not been yet accomplished. Following up on a previously published study (10.1016/j.stemcr.2022.05.014), the authors embark on a comprehensive study on the molecular and functional characterization of HSCs separated by CD49b – encoded by the *Itgb2* gene – now looking at age differences between young, adult and aged HSCs. This study demonstrates that although CD49b⁻ and CD49b⁺ HSCs have a similar transcriptome, there are significant epigenetic differences between these populations, and

these are deepened upon aging. Functionally, CD49b⁻ HSC demonstrate more “aging” traits such as myeloid-bias, early exhaustion upon serial transplantation.

Although the present study provides further relevant clues about the mechanisms whereby myeloid skewing arises, it misses on the opportunity to explore novel and different angles which could have shed further evidence of the relevance of CD49b’s function within the HSC compartment.

1. The present study has a major focus on epigenetic characterization, but the evidence that there are no major transcriptional changes between CD49b populations weakens the grounds for this pursue.

The reviewer is correct that our single cell RNA sequencing (scRNA-seq) data did not show any major transcriptional differences between the CD49b populations. This observation was initially reported in our previous publication where we used adult mice. In the current study, we have reproduced the finding using both juvenile and old mice. Despite transcriptional similarities, the CD49b HSC subsets exhibited distinct epigenetic profiles, which persisted through aging. This suggests that functionally diverse HSCs can be molecularly resolved by epigenetic analysis, but not by gene expression analysis. Our results further indicate that HSC subsets are epigenetically primed for HSC-specific behaviors, as shown by our transplantation experiments. The functional differences between CD49b subsets are supported by the enriched pathways identified through gene ontology and motif enrichment analyses. These results highlight the novel potential of epigenetic profiling in understanding the functional diversity of HSCs. We have described these findings in Results pages 13-14, lines 320-343, and in the Discussion page 17, lines 406-413; page 18, lines 427-432; 436-442.

2. Since the protagonist of the study is CD49b – a integrin – it is surprising that the authors do not further investigate the actual function in terms of adhesion/signaling properties. A reasonable hypothesis could be that CD49b populations have different integrin subunit composition in their membrane which in turn would yield different adhesive properties, potentially having a profound impact on the spatial localization of these cells in the BM and therefore on the HSC behaviors (doi.org/10.1016/j.stem.2013.06.015). Although the current manuscript has an impressive body of work, characterization of these population in the BM niche context would significantly strengthen the argument that CD49b is a key role player during the aging process, analogously to similar studies ([10.1016/j.stem.2010.11.030](https://doi.org/10.1016/j.stem.2010.11.030) and more recently [10.1038/s41467-020-20801-0](https://doi.org/10.1038/s41467-020-20801-0)).

In our study, we have used CD49b mainly as a marker to distinguish different HSC subtypes. Thus, we have not focused on investigating the functional role of CD49b in adhesion, migration, or localization of the CD49b subsets. However, the reviewer raises interesting and important points about the relationship between the role of CD49b and the BM niche. Still, we must acknowledge that not all phenotypic CD49b⁻ and CD49b⁺ cells are functional HSCs. In fact, prospective isolation of pure HSCs is not possible, which imposes limitations on any adhesion or migration experiments. The only valid assay for detecting and studying HSCs remains retroactive analysis through transplantation experiments. Consequently, addressing the reviewer’s request to specifically study adhesion and migration properties in *functional* CD49b⁻ and CD49b⁺ HSCs is very challenging, as there are no available assays that can definitely assess HSCs in a prospective manner. However, we appreciate the comments and have made efforts to partly address these by examining the expression of adhesion molecules in CD49b⁻ and CD49b⁺ subsets in juvenile, adult, and old mice. We have also performed *in vitro* migration assays, as well as analyzed the distribution of phenotypic CD49b⁻ and CD49b⁺ cells in different BM locations. Although these analyses do not directly address the role of CD49b in adhesion, migration, or homing of *functional* HSCs, they still give insights into the potential differences of the two

phenotypic CD49b⁻ and CD49b⁺ compartments (which include functional HSCs) and could provide the basis for further in-depth studies.

Taken together, I would consider this study for publication at Nat. Comms. with substantial revisions. Specifically, further studies focusing on the role of CD49b in terms of adhesion (e.g. in vivo or in vitro homing studies, BM niche localization of CD49b⁻ vs. CD49b⁺ populations).

In accordance with the reviewer's suggestions, we have included the following new experiments in the revised manuscript:

1. Distribution of phenotypic CD49b⁻ and CD49b⁺ HSCs in different BM locations. Supplementary Fig. 5a-c, Results page 8, lines 183-189, and Discussion page 16, lines 377-380.

The results showed comparable distribution of CD49b⁻ and CD49b⁺ HSCs in the central marrow, endosteal, and trabecular regions of steady-state mice, suggesting no selective preference for BM localization. Moreover, there was an increase of localization in the central marrow for both CD49b subsets. The data indicate that the differential functions of CD49b⁻ and CD49b⁺ HSCs are likely not dependent on their distribution in the BM.

2. *In vitro* migration analysis. Supplementary Fig. 5d-f, Results pages 8-9, lines 190-195, and Discussion page 16, lines 381-382.

Using a co-culture assay with mesenchymal stem cells, we assessed the adhesion and migration potential of phenotypic CD49b⁻ and CD49b⁺ HSCs. We observed an increased migration potential in CD49b⁺ HSCs in all age groups, but an age-related reduction in migration.

3. Gene expression analysis using KEGG adhesion molecule signature genes. Supplementary Fig. 6g, Results page 10, lines 227-229, Discussion page 16, line 381-382. Utilizing published data sets with signature genes for adhesion molecules, we could assess the gene expression differences between CD49b⁻ and CD49b⁺ HSCs, and across ages. We observed age-related differences in enrichment of adhesion molecules, consistent with the age-associated changes observed in migration potential.

4. Gene expression and FACS analysis of integrin beta 1, the binding partner of CD49b. Supplementary Fig. 6h, Results page 10, lines 229-230. We assessed the expression of integrin beta 1 by gene expression and by FACS, and observed no differences between subsets or by age, suggesting that CD49b, and not the binding partner, might contribute to the age-related changes in HSC migration.

Altogether, our new data indicate that phenotypic CD49b⁻ and CD49b⁺ HSCs locate in similar BM regions but have distinct migratory capacity, which is reduced in aging.

Beyond the aforementioned points:

Conceptual Comments:

3. It seems that the ratio between CD49b⁻ and CD49b⁺ is of essence. Thus, showing these groups separated by age is distracting, I would suggest replotting the data grouping based on CD49b expression rather than by age. Comparisons would be clearer. For example, Figure 1F seems to demonstrate that in old mice, CD49b⁺ incorporate more BrdU than CD49b⁻, but this is not clear.

We agree that comparative analysis between the CD49b⁻ and CD49b⁺ subsets is important. Hence, we have included not only age-related comparisons, but also graphs depicting differences between subsets in our Supplementary figures. In Supplementary Fig. 1c, we have graphically depicted the ratio between CD49b⁻ and CD49b⁺ within each age group (page 5, lines 97-99). Supplementary Fig. 1e shows cell cycle comparisons between the CD49b⁻ and CD49b⁺ subsets in juvenile, adult, and old age groups (page 5, lines 104-106). Lastly, Supplementary Fig. 1g

shows differences in BrdU incorporation between CD49b⁻ and CD49b⁺ subsets across age groups (page 5, lines 110-111).

4. In Figure 1, the authors assess frequencies and cells cycle stages of CD49b⁻ and CD49b⁺ within the LSK/SLAMhigh/CD34⁻ gate. However, it has been reported by different research groups (10.1182/blood-2008-12-192054 and doi.org/10.1073/pnas.1000834107) that SLAMhigh cells are myeloid-bias. One would predict that with the current gating strategy CD49⁺ cells which are not myeloid-bias and potentially CD150^{low}, yet still hold stem cell potential are being excluded for the analyses. Would it be any difference in terms of frequency or ration between CD49b⁻ and CD49b⁺?

The reviewer is correct that LSK CD34⁻CD150^{hi} cells have been reported to include myeloid-biased HSCs. However, the LSK CD34⁻CD150^{hi} compartment was still heterogeneous. We demonstrated in our previous publication that the CD49b marker could subfractionate the heterogeneous LSK CD48⁻CD34⁻CD150^{hi} cells into myeloid-biased enriched CD49b⁻ HSCs and lymphoid-biased enriched CD49b⁺ HSCs (Somuncular *et al.*, *Stem Cell Rep.*, 2022). The lymphoid-biased CD49b⁺ HSCs exhibited potent HSC potential, in contrast to LSK CD34⁻CD150^{-/low} (CD150^{-/low}) cells, which was reported previously to contain lymphoid-biased cells (Morita *et al.*, *J Exp Med.*, 2010; Kent *et al.*, *Blood.*, 2009). In our previous paper, we demonstrated that CD49b also subfractionates the LSK CD34⁻CD150^{int} (CD150^{int}) and CD150^{-/low} cells into CD49b⁻ and CD49b⁺ subfractions (Figure S1a in Somuncular *et al.*, *Stem Cell Rep.*, 2022). Through transplantation experiments, we showed that CD150^{int} and CD150^{-/low} subsets (without CD49b subfractionation) have reduced HSC potential compared to CD49b⁻ and CD49b⁺ HSCs. Therefore, in the current study, we only examined the age-related changes in CD49b⁻ and CD49b⁺ HSCs.

5. Still in Figure 1, it is demonstrated that there is a significant expansion in absolute number of both CD49b subsets upon aging, yet the authors conclude that the same populations become more quiescent, which seems a contradictory. How the authors explain these findings? Moreover, the authors should also contextualize their cell cycle findings in light with conflicting previous publications where authors demonstrate that HSC become more or less cycling upon aging.

It is correct that aging is associated with a dramatic expansion of both CD49b⁻ and CD49b⁺ cells. Our research, supported by cell cycle analysis, BrdU labeling experiments, and gene expression analysis, demonstrated that aged CD49b⁻ and CD49b⁺ cells exhibited a higher degree of quiescence and a lower proliferation rate compared to their younger counterparts. While these results may seem counterintuitive, they are consistent with a gradual expansion of the HSC compartment during aging. The expansion is likely due to symmetrical stem cell divisions in juvenile mice, as proposed in a recent publication (Arai *et al.*, *Cell Syst.*, 2020). Conversely, in old mice, symmetrical cell divisions generating two progenitors were more prevalent, leading to impaired HSC function. Despite this, the phenotypic HSC compartment in old mice remains relatively larger than in juvenile and adult mice. This observation suggests that the large HSC compartment may be maintained by limiting cell cycling, which would be consistent with our results. However, the cell cycle status of aged HSCs has been a subject of debate (Kovtonyuk *et al.*, *Front Immunol.*, 2016). This is likely due to variations in cell cycle analysis protocol and immunophenotype, which reflect differences in HSC purity. In the revised manuscript, we have included a discussion about the potential correlation between age-related HSC expansion and increased quiescence. We have also discussed the conflicting results concerning cell cycle status of aged HSCs (pages 16-17, lines 392-400).

6. In Figure 4 the authors performed scRNA-seq in different progenitor populations. Due to HSC pool heterogeneity, specially upon the aging process, it would be interesting to demonstrate how the author's data stand compared to different already published data sets (e.g.

10.1186/s12915-021-00955-z). The authors can also explore different previous findings (10.1038/nature25168) to further corroborate with their cell cycle findings. By comparing own data to previously published datasets would not only strengthen the author's argument but also would help to contextualize these findings.

We thank the reviewer for this suggestion. We have done further analysis to put our scRNA-seq data in the context of previous studies by downloading available data from studies analyzing similar populations as in our study with a focus on HSC identity and aging. Collectively, these studies analyzed stem- and progenitor cells from mice at ages corresponding to what we designated as adult (2-4 months) and old (1.5-2 years).

To corroborate the LT-HSC identity of the cells in our CD49b subsets, we used the scRNA-seq data from Rodriguez-Fraticelli *et al.*, *Nature*, 2018 (<https://www.ncbi.nlm.nih.gov/pmc/articles/PMC5884107/>). Assigning our HSCs to a cell type defined by FACS (in the Rodriguez-Fraticelli *et al.* study) based on expression similarity, we found that our HSCs were annotated as LT-HSCs (Supplementary Fig. 6b), supporting the stemness of the HSCs analyzed in our study. To further characterize potential age- and subset-related differences in lineage priming, we utilized the scRNA-seq data generated in Hérault *et al.*, *BMC Biol.*, 2021 (<https://pubmed.ncbi.nlm.nih.gov/33526011/>). This data was generated using a mixed population of Lin⁻Sca1⁺cKit⁺Flt3⁻ cells (which represent HSCs, MPP2, and MPP3) from adult or old mice. Labeling our cells with these annotations based on similarity in expression, we found that, independent of age or CD49b expression, most of our HSCs were annotated as non-primed (np)1, np2, or interferon response (ifn) expression clusters (Supplementary Fig. 6c). Overall, the data indicate that our HSCs lack significant expression of transcripts associated with mature lineages, as expected (Månsson *et al.*, *Immunity*, 2007).

Next, to compare our findings regarding HSC aging to previously published data, we downloaded counts tables from four other aging studies using scRNA-seq to compare adult and old LT-HSCs:

Grover *et al.*, *Nat. Commun.*, 2016: <https://pubmed.ncbi.nlm.nih.gov/27009448/>

Kowalczyk *et al.*, *Genome Res.*, 2015: <https://pubmed.ncbi.nlm.nih.gov/26430063/>

Kirschner *et al.*, *Cell Rep.*, 2017: <https://pubmed.ncbi.nlm.nih.gov/28538171/>

Mann *et al.*, *Cell Rep.*, 2018: <https://pubmed.ncbi.nlm.nih.gov/30540934/>

All data were integrated and displayed separately using UMAP (Supplementary Fig. 6e). Overall, we found that LT-HSCs segregated based on age group, independent of study, suggesting that the aging profile is consistent across studies. We also assessed how the significant age-related differences, reported in Supplementary Table 1, were expressed in these studies. We found that similar overall changes were observed across all studies (Supplementary Fig. 6f). This is consistent with the GSEA enrichment observed using the aging HSC signature defined by Svendsen *et al.*, *Blood*, 2021 (<https://pubmed.ncbi.nlm.nih.gov/33876187/>) (Fig. 4c).

To our knowledge, there are no prior studies comparing juvenile, adult, and old LT-HSCs using scRNA-seq. However, Li *et al.*, *Cell Stem Cell.*, 2020 have compared fetal, neonatal, and adult HSCs. Using their data set, we confirmed that our juvenile HSCs did not express a fetal HSC signature but were more similar to adult HSCs (Supplementary Fig. 6d).

We agree with the reviewer that addressing cell cycle in our scRNA-seq data could further corroborate our cell cycle and proliferation findings from Ki-67 and BrdU-incorporation assays. To our understanding, Rodriguez-Fraticelli *et al.*, *Nature.*, 2018, referenced by the reviewer, does not seem to address HSC cell cycle regulation in depth. We instead used the HSC quiescence and proliferation gene sets defined by Venezia *et al.*, *PLoS Biol.*, 2004 (<https://pubmed.ncbi.nlm.nih.gov/15459755/>) to calculate a quiescence and proliferation score for each cell of our scRNA-seq data. As expected, we found that our HSC subsets had a significantly lower proliferation score and significantly higher quiescence score compared to both LMPPs and GMPs (Fig. 4d). We also observed a significantly higher proliferation score and lower quiescence score in juvenile HSCs subsets, compared to their adult and old counterparts (Fig. 4e). These

analyses corroborate our experimental findings demonstrating that both CD49b subsets become progressively more quiescent with age (Fig. 1d-f).

We have included these results in Fig. 4d and Supplementary Fig. 6b-f, and described the findings in the revised manuscript page 10, lines 221-223; page 9, lines 208-211; page 9, lines 216-217; page 10, lines 226-227.

7. For the ATAC-seq data which is described in Figures 5 and 6, how does the data correlates with previously published ATAC-seq from young and aged mice? doi.org/10.1038/s41467-022-30440-2

To compare our data to that from Itokawa *et al.*, *Nat. Commun.*, 2022 (<https://pubmed.ncbi.nlm.nih.gov/35577813/>), we downloaded ATAC-seq data from the populations matching our study (HSC, MPP4 = LMPP, and GMP), processed the data using our pipeline, and integrated the two datasets using batch correction to account for methodological differences between the studies. The cells analyzed in Itokawa *et al.* were isolated from either 10-week- or 20-month-old mice, which correspond to our adult (2-4 months) and old (1.5-2 years) mice. Performing principal component analysis (PCA), we found that samples clustered together based on cell type across both studies (Supplementary Fig. 8b). The PCA also indicated age-related separation (Supplementary Fig. 8b). Hierarchical clustering based on Spearman correlation between the samples confirmed the age-related separation (Supplementary Fig. 8c). We found that regions with decreased accessibility in aging (Fig. 5c, cluster 1) also showed decreased accessibility in the Itokawa *et al.* data (Supplementary Fig. 8e). Conversely, regions with increased accessibility in aging (Fig. 5c, clusters 2 and 3) showed intermediate accessibility in adult samples from both studies, which then further increased with age (Supplementary Fig. 8e). In addition, consistent with the HSC data in Itokawa *et al.* representing a mixture of CD49b⁺ and CD49b⁻ HSCs, we found that the accessibility differences between CD49b⁺ and CD49b⁻ HSCs from old mice had intermediate signals in the Itokawa *et al.* HSCs (Supplementary Fig. 9b). Overall, we conclude that the data from the two studies correlate well, which supports the validity of our findings.

We have included these results in our manuscript as Supplementary Figs. 8b,c,e,9b, and described the findings in the revised manuscript page 11, lines 258-259; page 11, lines 266-268; page 13, lines 310-311.

Minor comments:

8. In Line 84 replace “development” as this seems to indicate that the study is about the early life events, not as in lifespan.

We have now rephrased the sentence (page 4, lines 79-80).

9. The references 12 and 23 do not mention CD49b in line 77, please reframe it.

We thank the reviewer for pointing out the inconsistencies in the references. We have removed references not mentioning CD49b to avoid confusion.

10. In Figure 2C, the chimerism plot should not be log-transformed as it's hard to see potential differences.

We presented the chimerism plot in log scale to better compare the reconstitution levels among the different age groups. However, we have now represented the data in linear scale to show variations among individual mice more clearly (see Fig. 2c).

Reviewer #3:

The authors Somuncular, Hauenstein, Su et al. describe the functionality of two distinct HSC

subtypes (CD49b[±]) during aging and how aging alters their lineage commitment. The authors clearly state transcriptional similarities but emphasize differences on the epigenetic level between young and old HSC subtypes. Their results demonstrate how myeloid lineage bias could be affected by lineage specific transcription factors and higher accessibilities for their corresponding binding motifs with age.

The presented data is of high quality, obtained with appropriate techniques and supports the authors' conclusions. Some of the provided figures are missing statistical analysis, which will be commented on in the section below. Since the impact of aging on HSC functionality and diversity is broadly discussed and has a wide impact on the field of hematopoietic aging, the community highly profits from the presented results.

Said that, this reviewer hopes that the following comments help to improve the clarity of this manuscript.

1. The authors describe an increase of CD49b[±] HSCs with age (Fig. 1B,C). The overall contribution of CD49b[±] cells to the composition of the aging HSC pool would be easier to judge if the percentage of CD49b[±] cells within the HSC pool would be graphically displayed between young, adult and old mice.

In Supplementary Fig. 1c, we are showing the percentage of CD49b⁻ and CD49b⁺ within the HSC pool of juvenile, adult, and old mice. The distribution pattern was generally the same across ages, however, the number of CD49b⁻ cells was significantly higher in both juvenile and old mice, with the biggest difference observed in the old age group. We have updated the revised manuscript on page 5, lines 96-99.

2. While the authors clearly show that the cycling activity of the HSC subtypes changes with age, this reviewer is missing a statement of the authors discussing the high quiescence (Fig. 1D,E,F) but reduced HSC-score (Fig. 4D) of old CD49b⁺ cells regarding their self-renewal potential also considering previously published work describing aged HSCs to be frequently cycling (e.g. Morrison et al., 1996).

Our data using cell cycle and BrdU labeling experiments demonstrated that aged CD49b⁻ and CD49b⁺ cells exhibited a higher degree of quiescence and a lower proliferation rate compared to their juvenile counterparts (Fig. 1d,f). However, we still observed that CD49b⁺ cells were significantly less quiescent and more proliferative than the CD49b⁻ subsets (Supplementary Fig. 1e,g). These results were corroborated with the reduced engraftment potential of CD49b⁺ cells in secondary transplantation experiments (Fig. 3d), and the significantly lower HSC-score in adult and old age groups. These results are compatible with CD49b⁻ HSCs harboring the highest self-renewal potential. We were unable to see any significant difference in HSC-score in the juvenile group despite functional differences, suggesting that HSC-score may not be able to detect all self-renewal differences. We have updated the results and discussion sections to include an improved description and clarification about these data on page 10, lines 237-242, page 17, lines 400-402.

Our cell cycle and cell proliferation results contrast previous data by Morrison *et al.*, *Nat Med.*, 1996. The discrepancy is likely due to differences in cell cycle analysis protocol and immunophenotype, which reflect different levels of HSC purity. While the Morrison *et al.* publication analyzed the stem- and progenitor population, we have assessed highly enriched HSC subsets. We have updated the discussion with the conflicting results concerning cell cycle status in aged HSCs (pages 16-17, lines 392-405).

3. The provided data for in vitro functionality of young CD49b⁺ HSCs, which have been described to have lymphoid bias, show lower B cell frequencies compared to myeloid frequencies (Fig. 2A). To strengthen the aspect of CD49b⁺ being lymphoid-biased HSC subsets, the manuscript would benefit from a comment on this lack of lymphoid over myeloid differentiation potential for young CD49b⁺ HSCs. In case the provided format of data display is misleading and indeed the lymphoid differentiation is significantly increased, this reviewer

would kindly ask the authors to rearrange the data display and to add necessary statistical comparisons.

The reviewer is correct in observing that there is a lower B cell frequency compared to myeloid frequency in the OP9 co-cultures. This is seen in both CD49b subsets and across age, and reflect the ability of the OP9 cells to support and promote myeloid cells more efficiently than B cells when culturing HSCs (Somuncular *et al.*, *Stem Cell Rep.*, 2022; Månsson *et al.*, *Immunity.*, 2007). Consequently, we have used the OP9 co-culture system to assess the differentiation abilities of CD49b HSCs to generate lymphoid and myeloid cells *in vitro*, and used transplantation experiments to examine the lineage bias properties. As supported by Fig. 2e and Supplementary Fig. 3c, CD49b⁺ HSCs preferentially produce lymphoid cells over myeloid cells, in contrast to CD49b⁻ HSCs. These observations are also corroborated by the enrichment of lymphoid-biased HSCs in CD49b⁺ subsets in both juvenile and adult age groups (Fig. 2f). We have updated the results (page 6, lines 119-120), with a clarification of the OP9 co-culture system and added a description of the lymphoid preference in CD49b⁺ HSCs, Supplementary Fig. 3c page 6, lines 129-132.

4. The authors further transplant CD49b^{+/-} cells to follow their reconstitution potential (Fig. 2D) but statistical analysis showing significant changes for the reconstitution of the myeloid vs. lymphoid compartment is missing and should be included to support their findings.

Our main purpose in Fig. 2d was to demonstrate the multipotent differentiation abilities of the CD49b⁻ and CD49b⁺ HSCs throughout aging, as well as to show the dynamic repopulation changes over time. These data showed that both CD49b subsets, in all age groups, were long-term reconstituting cells. On the other hand, Fig. 2e showed changes in the lineage distribution of individual mice, which suggested that the relative myeloid contribution was increased in aged mice, in both CD49b subsets. We agree with the reviewer that it could be helpful to view the total repopulation level of the myeloid and lymphoid compartment to complement Fig. 2d. We have therefore included an additional figure (Supplementary Fig. 3c), showing the myeloid vs lymphoid (B, T, and NK cells combined) reconstituted cells in CD49b⁻ or CD49b⁺ transplanted animals, 5-6 months post-transplantation (Supplementary Fig. 3c, page 6, lines 129-132). The results demonstrated that the repopulation level of lymphoid cells is significantly higher than that of myeloid cells in CD49b⁺ transplanted mice, as expected, while the lymphoid and myeloid repopulation level in CD49b⁻ transplanted mice were comparable.

5. The authors nicely show the reconstitution of the myeloid vs HSC compartment by CD49b^{+/-} cells in elegant secondary transplantations (Fig. 3C). However, this reviewer is missing a statistical comparison to judge the outcome of this experiment. As the data is presented, the authors show higher reconstitution of the HSC compartment by old CD49b⁺ HSCs compared to young CD49b⁺ HSCs and equal reconstitution of the myeloid and HSC compartment by CD49b⁻ old and young HSCs. This reviewer is missing a critical discussion with previously published literature describing reduced reconstitution by old HSCs (e.g. Dykstra *et al.*, 2011, Morrison *et al.*, 1996, Poscablo *et al.*, 2021).

We thank the reviewer for pointing out a critical point. Given that mice were transplanted with different cell numbers from juvenile, adult, and old mice, it is not possible to compare repopulation levels across ages. However, the repopulation in primary transplantation were at similar level in all age groups, despite transplanting 20 times more old HSCs (Fig. 2c), which align with old HSCs harboring reduced engraftment potential (Dykstra *et al.*, *J Exp Med.*, 2010, Morrison *et al.*, *Nat Med.*, 1996). Furthermore, we have also included a new figure (Fig. 3d) to show the donor repopulation level in secondary transplanted mice. Interestingly, while the level of reconstitution was similar in juvenile and old groups, as observed in primary transplantation (Fig. 2c), CD49b⁺ HSCs from both age groups had significantly lower repopulation than the corresponding CD49b⁻ HSCs in secondary transplantation (Fig. 3d). We have now also separated the graphs reporting

the number of repopulated mice in myeloid cells or HSCs, according to age group, to avoid confusion (see updated Fig. 3a and Fig. 3c). These data demonstrate that in addition to reduced engraftment potential in CD49b⁺ HSCs (Fig. 3d), the number of mice exhibiting myeloid or HSC repopulation is also lower (Fig. 3a,c). Altogether, the results are compatible with CD49b⁻ HSCs harboring the highest self-renewal potential. We have added the new data on page 8, lines 172-179, and discussed the results on page 17, lines 400-402.

6. Statistically comparing the HSC-score (Fig. 4D) between young and old CD49b^{+/-} would improve the message of this figure.

In the revised manuscript, we have now added an additional figure showing the HSC-score comparison of CD49b⁻ or CD49b⁺ between age groups (Supplementary Fig. 6i, page 10, lines 241-242). However, we did not observe any consistent age-related changes, suggesting that HSC-score may not be able to detect all self-renewal differences.

7. The connection of higher chromatin accessibility in old HSC to myeloid differentiation is clearly demonstrated (Fig. 5). The meaning of this outcome should be proven by including absolute numbers of myeloid cells reconstituted by CD49b⁺ vs CD49b⁻ young vs old donor HSCs in the peripheral blood after transplantation.

Given that mice were transplanted with different cell numbers from juvenile, adult, and old mice, it is not possible to compare repopulation levels across ages. Instead, we have presented the data as lymphoid to myeloid (L/M) ratio to visualize the relative differences. As clearly shown, the L/M ratio is decreased with age, supporting that myeloid differentiation is increased in old mice compared to juvenile mice (Supplementary Fig. 8g, page 12, lines 279-281).

8. The authors convincingly show a decrease in PU.1 expression in old HSCs (extended Fig. 7). This reviewer is missing a comment on previously published work showing how high PU.1 levels instruct myeloid differentiation (e.g. Nerlov and Graf, 1998) which might be counterintuitive regarding the described myeloid bias of old HSCs in this manuscript. This is especially interesting, as the authors present the highest motif enrichment for PU.1 binding sites in young lymphoid biased CD49b⁺ cells (Fig. 6G).

We agree that it is important to discuss previously published work about PU.1 and its correlation with the age-related decrease of PU.1 binding sites. Notably, previous publications have demonstrated that high PU.1 promotes myelopoiesis whereas low PU.1 is associated with lymphoid development (Nerlov & Graf., *Genes Dev.*, 1998). However, interestingly, mice defective in PU.1 demonstrated an increase in granulocytic cells and increased risks of leukemia development (Dakic *et al.*, *J Exp Med.*, 2005; Metcalf *et al.*, *Proc Natl Acad Sci.*, 2006), which align with the aging phenotype. In the revised manuscript, we have added a discussion about PU.1 on page 17, line 415; page 18, lines 430-435, in accordance with the reviewer's suggestion.

9. As a minor comment for extended figure 3C, this reviewer is missing the legend explaining the color code.

We thank the reviewer for noticing the mistake. We have now updated the figure with legends included.

This reviewer highly recommends including missing statistical comparisons as well as a critical discussion about possible controversies between previously published work and new data provided in this manuscript to strengthen the important outcome of this study.

We appreciate the comment and have added missing statistical analyses throughout the revised manuscript. Furthermore, we have included a discussion about controversies between published work and our data. This includes a discussion about the age-related expansion of HSC compartment but increased quiescence with age (pages 16-17, lines 392-400), the cell cycle

status of HSCs in aged mice (page 16-17, lines 392-400), and the downregulation of PU.1 with age (page 17, line 415; page 18, lines 430-435).

REVIEWERS' COMMENTS

Reviewer #1 (Remarks to the Author):

The manuscript 'Aging is associated with functional and molecular changes in distinct hematopoietic stem cell subsets' by Somuncular et al. investigates by how different subtypes of murine HSCs, which are separated by CD49b, change upon aging. The authors provide a revised version of the manuscript. The manuscript remains organized in a logical flow and the experimental set-ups are sound. They provide additional confirmatory data requested primarily by reviewer 1 and 2. So the manuscript continues to provide solid descriptive data.

This reviewer remains skeptical on the novelty of the data, as listed in the review of the initial version of the manuscript. In the opinion of this reviewer, it continues to fall short on providing more novel approaches or additional mechanistic data that at least in the view of this reviewer would set this manuscript significantly apart from the information provided by other published studies. This reviewer does further not agree with the statement of the authors that mechanisms of aging of HSCs have not been identified. There are now multiple examples in the literature that might help to structure their data. This reviewer agrees with the authors that their data extend current concepts, and that their data align with multiple studies, while their approach might, similar to other published studies, likely identify new targets. Mechanistical approach though remains absent from their data, as already commented before. At the level of Nature Communications, such data would provide the novelty that goes beyond extending and alignment. These comments align with comments of reviewer 2. S/he was looking for information of localization relative to known niches as novel information that would significantly add to the manuscript. In conclusion, within the data and thus also within the manuscript, no new principles or techniques are reported and there are no mechanistical studies. The data do not really provide new approaches for intervention as claimed in the discussion. In the view of this reviewer, it remains descriptive work, and the results primarily reflect/confirm/extend, in a solid fashion, established data on HSC aging.

Reviewer #2 (Remarks to the Author):

In this new and revised version of the manuscript entitled "Aging is associated with functional and molecular changes in distinct hematopoietic stem cell subsets", the authors have added novel and significant experimental data, which further assist on placing their findings within the context of the HSC aging field.

In the current format, different aspects of HSC aging are now clearly defined. Specifically, major aspects which were raised were properly address. Specifically, this reviewer much appreciates the

addition of the finding that CD49b- and CD49b+ HSCs do not localize differently within the BM, but do have some migratory differences. Secondly, the authors now have also placed their cell cycle findings into context of previous contradictory findings, further suggesting that symmetrical and asymmetrical HSC division play an important role and it is context-dependent.

Taken together, it is this reviewer's opinion that the current manuscript has much improved and it would be a good asset into the HSC aging field. Therefore, I would recommend its acceptance.

Reviewer #3 (Remarks to the Author):

In the revised manuscript by Su, Hauenstein, Somuncular et al. the authors thoroughly updated critical points and figure parts that now strengthen the statement of functional and molecular changes in CD49b subsets within the HSC pool during aging. This referee would like to thank the authors for additional critical discussion of conflicting data to clarify the outcome, challenges and perspectives of their study.